# StyleGAN knows Normal, Depth, Albedo, and More

**Anand Bhattad**    **Daniel McKee**    **Derek Hoiem**    **D.A. Forsyth**
University of Illinois Urbana Champaign
https://anandbhattad.github.io/stylegan_knows/

| (a) Image | (b) Normal | (c) Depth | (d) Albedo | (e) Shading | (f) Segment |
|---|---|---|---|---|---|

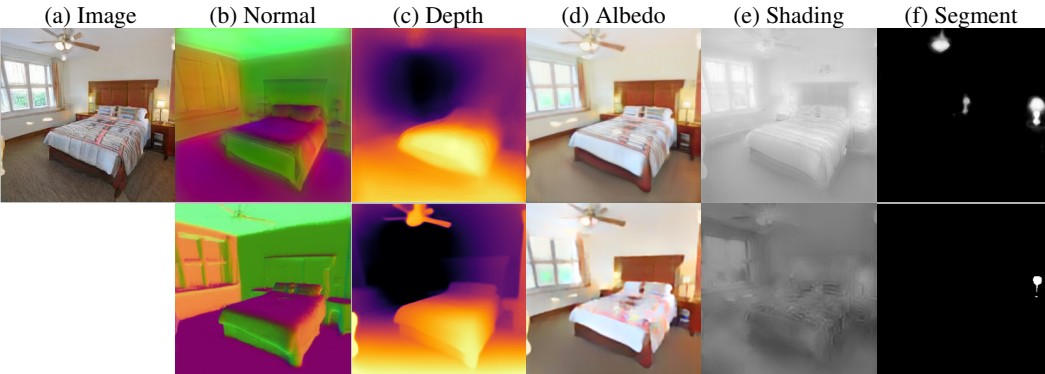

Figure 1: StyleGAN has easily accessible and accurate representations of intrinsic images, without ever having seen an intrinsic image. Simply by finding an appropriate offset to the latent variables for each type, we make StyleGAN reveal intrinsic images of many types for a synthesized image (a), including: surface normal (b), depth maps (c), albedo (d), shading (e), segmentation (f). No new weight learning or fine-tuning is required. **Top row:** shows StyleGAN intrinsics; **bottom row** those predicted by SOTA predictors [28, 10, 20, 18]. Note that StyleGAN "knows" where bedside and other lamps are better than a SOTA segmenter [18] does (it should; it put them there!) and that StyleGAN "knows" fine detail in normal maps (around bedside lamp) that is hard for current methods to predict.

## Abstract

Intrinsic images, in the original sense, are image-like maps of scene properties like depth, normal, albedo or shading. This paper demonstrates that StyleGAN can easily be induced to produce intrinsic images. Our procedure is straightforward. We show that, if StyleGAN produces $G(w)$ from latent $w$, then for each type of intrinsic image, there is a fixed offset $d_c$ so that $G(w + d_c)$ is that type of intrinsic image for $G(w)$. Here $d_c$ is *independent of* $w$. The StyleGAN we used was pretrained by others, so this property is not some accident of our training regime. We show that there are image transformations StyleGAN will *not* produce in this fashion, so StyleGAN is not a generic image regression engine.

It is conceptually exciting that an image generator should "know" and represent intrinsic images. There may also be practical advantages to using a generative model to produce intrinsic images. The intrinsic images obtained from StyleGAN compare well both qualitatively and quantitatively with those obtained by using SOTA image regression techniques; but StyleGAN's intrinsic images are robust to relighting effects, unlike SOTA methods.

37th Conference on Neural Information Processing Systems (NeurIPS 2023).

# 1 Introduction

Barrow and Tenenbaum, in an immensely influential paper of 1978, defined the term "intrinsic image" as "characteristics – such as range, orientation, reflectance and incident illumination – of the surface element visible at each point of the image" [6]. Maps of such properties as (at least) depth, normal, albedo, and shading form different types of intrinsic images. The importance of the idea is recognized in computer vision – where one attempts to recover intrinsics from images – and in computer graphics – where these and other properties are used to generate images using models rooted in physics. But are these representations in some sense natural? In this paper, we show that a marquee generative model – StyleGAN – has easily accessible internal representations of many types of intrinsic images (Figure 1), without ever having seen intrinsics in training, suggesting that they are.

We choose StyleGAN [30, 31, 29] as a representative generative model because it is known for synthesizing visually pleasing images and there is a well-established literature on the control of StyleGAN [65, 11, 75, 14, 53, 57]. Our procedure echoes this literature. We search for offsets to the latent variables used by StyleGAN, such that those offsets produce the desired type of intrinsic image (Section 4; Figure 2). We use a pre-trained StyleGAN which has never seen an intrinsic image in training, and a control experiment confirms that StyleGAN is not a generic image regressor. All this suggests that the internal representations are not "accidental" – likely, StyleGAN can produce intrinsic images because (a) their spatial statistics are strongly linked to those of image pixels and (b) they are useful in rendering images.

There may be practical consequences. As Section 5 shows, the intrinsic images recovered compare very well to those produced by robust image regression methods [28, 10, 18, 20], both qualitatively and quantitatively. But StyleGAN produces intrinsic images that are robust to changes in lighting conditions, whereas current SOTA methods are not. Further, our method does not need to be shown many examples (image, intrinsic image) pairs. These practical consequences rest on being able to produce intrinsic images for real (rather than generated) images, which we cannot currently do. Current SOTA GAN inversion methods (eg [3, 47, 12, 61]) do not preserve the parametrization of the latent space, so directions produced by our search do not reliably produce the correct intrinsic *for GAN inverted images*. As GAN inversion methods become more accurate, we expect that generative models can be turned into generic intrinsic image methods. Our contributions are:

- Demonstrating that StyleGAN has accessible internal representations of intrinsic scene properties such as normals, depth, albedo, shading, and segmentation without having seen them in training.

- Describing a simple, effective, and generalizable method, that requires no additional learning or fine-tuning, for extracting these representations using StyleGAN's latent codes.

- Showing that intrinsic images extracted from StyleGAN compare well with those produced by SOTA methods, and are robust to lighting changes, unlike SOTA methods.

# 2 Related Work

**Generative Models:** Various generative models, such as Variational Autoencoders (VAEs)[33], Generative Adversarial Networks (GANs)[21], Autoregressive models [58], and Diffusion Models [16], have been developed. These models have made significant improvements in training and output quality through novel loss functions and stability enhancements despite initial challenges with blurry outputs [49, 45, 30, 48, 27]. In this work, we focus on StyleGAN [30, 31, 29] due to its exceptional ability to manipulate disentangled style representations with ease. We anticipate that analogous discoveries will be made for other generative models in the future.

**Editing in Generative Models:** A variety of editing techniques allow for targeted modifications to generative models' output. One prominent example is StyleGAN editing [53, 60, 62, 52, 74, 15, 46, 14, 11, 12, 2, 55], which allows for precise alterations to the synthesized images. Similarly, a handful of editing methods have emerged for autoregressive and diffusion models [23, 5, 13, 69, 39, 32, 37, 59].

In the context of StyleGAN editing, there exist several approaches such as additive perturbations to latents [65, 11, 75, 53, 57], affine transformation on latents [62, 26], layer-wise editing [64], activation-based editing [8, 7, 15], and joint modeling of images and labeled attributes [52, 71, 34, 35]. These methods facilitate nuanced and specific changes to the images.

In our study, we adopt straightforward and simplest additive perturbations to latents, rather than learning new transformations or engineering-specific layer modifications. By searching for small latent code perturbations to be applied across all layers, we allow the model to learn and modulate how these new latent representations influence the generated output. This process minimizes the need for intricate layer-wise manipulation while still achieving desired edits and providing valuable insights into the model's internal latent structure.

**Discriminative Tasks with Generative Models:** An emerging trend involves leveraging generative models for discriminative tasks. Examples of such work include Labels4Free [1], GenRep [24], Dataset GAN [71], SemanticGAN [34] RGBD-GAN [40], DepthGAN [54], MGM [4], ODISE [63], ImageNet-SD [50], and VPD [72]. These approaches either use generated images to improve downstream discriminative models or fine-tune the original generative model for a new task or learn new layers or learn new decoders to produce desired scene property outputs for various tasks.

In contrast to these methods, our approach eschews fine-tuning, learning new layers, or learning additional decoders. Instead, we directly explore the latent space within a pretrained generative model to identify latent codes capable of predicting desired scene property maps. This process not only simplifies the task of utilizing generative models for discriminative tasks but also reveals their inherent ability to produce informative outputs without extensive modification or additional training.

**Intrinsic Images:** Intrinsic images were introduced by Barrow and Tenenbaum [6]. Intrinsic image prediction is often assumed to mean albedo prediction, but the original concept "characteristics ... of the surface element visible at each point of the image" ([6], abstract) explicitly included depth, normals, and shading; it extends quite naturally to semantic segmentation maps too. Albedo and shading (where supervised data is hard to find) tend to be studied apart from depth, normals, and semantic segmentation (where supervised data is quite easily acquired). Albedo and shading estimation methods have a long history. As the recent review in [20] shows, methods involving little or no learning have remained competitive until relatively recently; significant recent methods based on learning include [25, 67, 36, 20]. Learned methods are dominant for depth estimation, normal estimation, and semantic segmentation. Competitive recent methods [17, 28, 44, 10, 19] require substantial labeled training data and numerous augmentations.

**What is known about what StyleGAN knows:** Various papers have investigated what StyleGAN knows, starting with good evidence that StyleGAN "knows" 3D information about faces [41, 70], enough to support editing [43, 42, 56]. Searching offsets (as we do) yields directions that relight synthesized images [11]. In contrast, we show that StyleGAN has easily accessible representations of natural intrinsic images, *without ever having seen an intrinsic image of any kind*.

## 3 Background

### 3.1 StyleGAN

StyleGAN [30, 29] uses two components: a mapping network and a synthesis network. The mapping network maps a latent vector $\mathbf{z}$ to an intermediate space $\mathbf{w}$. The synthesis network takes $\mathbf{w}$, and generates an image $\mathbf{x}$, modulating style using adaptive instance normalization (AdaIN) [22]. Stochastic variation is introduced at each layer by adding noise scaled by learned factors. The architecture of StyleGAN can be summarized by the following equations:

$$\mathbf{w} = f(\mathbf{z})$$
$$\mathbf{x} = g(\mathbf{w}, \mathbf{n})$$

where $f$ is the mapping network, $g$ is the synthesis network, and $\mathbf{n}$ is a noise vector.

### 3.2 Manipulating StyleGAN

StyleGAN's intermediate latent code, denoted as $\mathbf{w}$, dictates the style of the image generated. The $\mathbf{w}^+$ space is a more detailed version of the $\mathbf{w}$ space, where a unique $\mathbf{w}$ vector is provided to each layer of the synthesis network [62]. This allows for more fine-grained control over the generated image at varying levels of detail.

Editing in StyleGAN can be achieved by manipulating these $\mathbf{w}$ vectors in the $\mathbf{w}^+$ space. We do this by identifying a new latent code $\mathbf{w}^{+\prime}$ that is close to the original $\mathbf{w}^+$ but also satisfies a specific

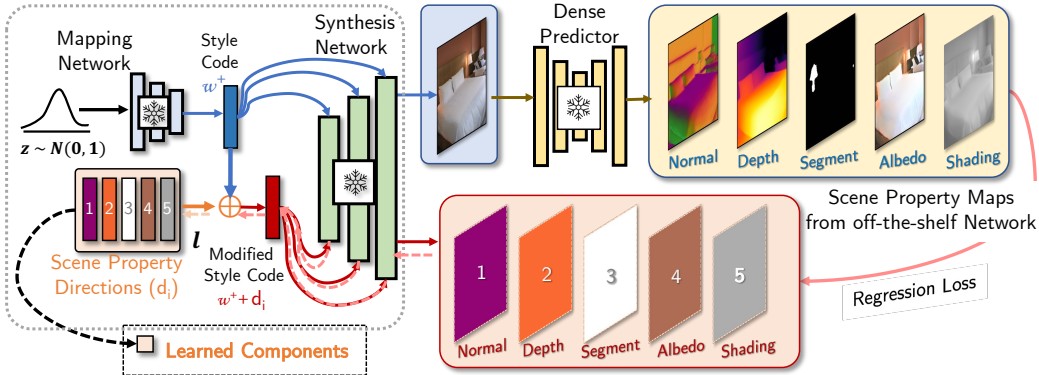

(a) Searching for directions ($d_i$) corresponding to different scene intrinsic map in Latent ($w^+$) Space

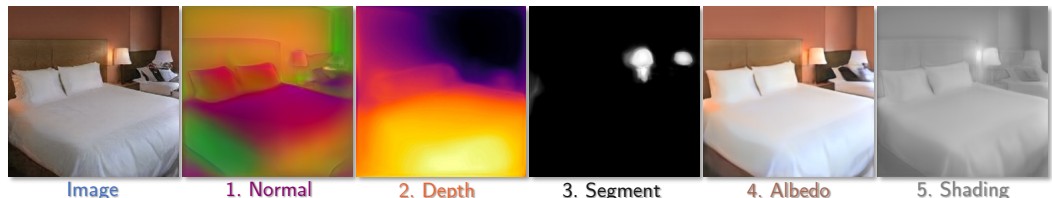

| Image | 1. Normal | 2. Depth | 3. Segment | 4. Albedo | 5. Shading |

(b) Final scene property maps generated from StyleGAN after search completion

Figure 2: **Searching for scene intrinsic offsets.** We demonstrate that a StyleGAN trained only to generate images also encodes accessible scene property maps. We use a simple way to extract these scene property maps. Our approach explores latent directions ($d$) or offsets within a pretrained StyleGAN's space, which, when combined with the model's style codes ($w^+$), generates surface normal predictions, depth predictions, segmentation, albedo predictions, and shading predictions. Importantly, our approach does not require any additional fine-tuning or parameter changes to the original StyleGAN model. Note the StyleGAN model was trained to generate natural scene-like images and was never exposed to scene property maps during training. We use off-the-shelf, state-of-the-art dense prediction networks, only to guide this exploration. The discovery of these scene property latents offers valuable insights into how StyleGAN produces semantically consistent images.

editing constraint $\mathbf{c}$ [65, 11, 75, 53, 57]. This problem can be formulated as:

$$\mathbf{w}^{+'} = \mathbf{w}^+ + \mathbf{d}(\mathbf{c}),$$

where $\mathbf{d}(\mathbf{c})$ computes a perturbation to $\mathbf{w}$ based on $\mathbf{c}$. $\mathbf{d}(\mathbf{c})$ can be found using methods like gradient descent. The edited image is then generated from the new latent code $\mathbf{w}^{+'}$.

## 4 Searching for Intrinsic Offsets

Our approach is illustrated in Figure 2. We directly search for specific perturbations or offsets, denoted as $\mathbf{d}(\mathbf{c})$, which when added to the intermediate latent code $\mathbf{w}^+$, i.e., $\mathbf{w}^{+'} = \mathbf{w}^+ + \mathbf{d}(\mathbf{c})$, yield the desired intrinsic scene properties. Different perturbations $d(c)$ are used for generating various intrinsic images such as normals, depth, albedo, shading, and segmentation masks. To search for these offsets, we utilize off-the-shelf pretrained networks from Omnidata-v2 [28] for surface normals, Zoe-depth [10] for depth, EVA-2 [18] for semantic segmentation, and Paradigms for intrinsic image decomposition [20] to compute the desired scene properties for the generated image $\mathbf{x} = G(\mathbf{z})$, We employ a Huber-loss (smooth L1-Loss) to measure the difference between generated intrinsic and off-the-shelf network's predicted intrinsics. Formally, we solve the following optimization problem:

$$\mathbf{w}^{+'} = \arg \min_{\mathbf{d}(\mathbf{c})} \mathrm{L}_{\mathrm{Huber}}(P(\mathbf{x}), \mathbf{x}'),$$

where $P$ is a function that computes the scene property map from an image. By solving this problem, we obtain a latent code $\mathbf{w}'$ capable of generating an image $\mathbf{x}'$ that are scene properties as $\mathbf{x}$. We also note that task-specific losses, such as scale-invariant loss in depth or classification loss in

segmentation, do not significantly contribute to the overall performance. This observation suggests that our approach is robust and generalizable, opening the door for additional applications in other variants of scene property prediction. We found that incorporating task-specific losses, such as angular loss for normal prediction, slightly improves the quality of predicted intrinsic images.

## 5 Accuracy of StyleGAN Intrinsics

For the intrinsic of type $\mathbf{c}$, we search for a direction $\mathbf{d}(\mathbf{c})$ using randomly selected images generated by StyleGAN using $\mathbf{w}^+$ and a reference intrinsic image method. Although approximately 200 images can successfully identify the required directions, we used 2000 unique scenes or generated images in our experiments. Note that this is not a scene-by-scene search. A direction is applicable to all StyleGAN images once it is found. For each generated image, we obtain target intrinsics using a SOTA reference network. We then synthesize intrinsic images from StyleGAN using the formula $\mathbf{w}^+ + \mathbf{d}(c)$. These synthesized intrinsic images are subsequently compared to those produced by leading regression methods, using standard evaluation metrics.

Importantly, the StyleGAN model used in our work has never been trained on any intrinsic images. We use a pretrained model from Yu et al. [66] and that remains unaltered during the entire latent search process. Off-the-shelf networks are exclusively utilized for latent discovery, not for any part of the StyleGAN training. Overall time to find one intrinsic image direction is less than 2 minutes on an A40 GPU. In total, less than 24 hours of a single A40 GPU were required for the final reported experiments, and less than 200 hours of a single A40 GPU were required from ideation to final experiments.

We have no ground truth. We evaluate by generating a set of images and their intrinsics using StyleGAN. For these images, we compute intrinsics using the reference SOTA method and treat the results (reference intrinsics) as ground truth. We then compute metrics comparing StyleGAN intrinsics with these results; if these metrics are good, the intrinsics produced by StyleGAN compare well with those produced by SOTA methods. In some cases, we are able to calibrate. We do so by obtaining the intrinsics of the generated images using other SOTA methods (calibration metrics) and comparing these with the reference intrinsics.

**Surface Normals:** We rely on Omnidata-v2 [28] inferred normals as a reference, comparing both the L1 and angular errors of normal predictions made by calibration methods (from [17, 68]) and StyleGAN. As shown in Table 1, the surface normals generated by StyleGAN are somewhat less accurate quantitatively in comparison. A visual comparison is provided in Figure 3 and Figure 7.

**Depth:** We utilize ZoeDepth [10] inferred depth as a reference, comparing the L1 error of depth predictions made by calibration methods (from [28, 17, 68]) and StyleGAN. Interestingly, depth predictions made by StyleGAN surpass these methods in performance, as shown in Table 1. A visual comparison is available in Figure 4 and Figure 8.

**Albedo and Shading:** We use a recent SOTA, self-supervised image decomposition model [20] on the IIW dataset [9] to guide our search for albedo and shading latents. This involves conducting independent searches for latents that align with albedo and shading, guided by the regression model.

Table 1: **Quantitative Comparison of Normal and Depth Estimation.** We use Omnidata-v2 [28] and ZoeDepth [10] as pseudo ground truth when comparing for surface normals and depth respectively. StyleGAN model performs slightly worse on normals and slightly better on depth despite never having been exposed to any normal or depth data, operating without any supervision, and achieving this task in a zero-shot manner. It's important to note that our use of pre-trained normals or depth only serves to guide StyleGAN to find these latent codes that correspond to them. There is no learning of network weights involved, hence the performance is effectively zero-shot.

| Models | #Parameters | Normals | | Depth |
| --- | --- | --- | --- | --- |
| | Normals / Depth | L1 $\downarrow$ | Angular Error $\downarrow$ | L1 $\downarrow$ |
| Omnidata-v2 [28] | 123.1M | – | – | 0.1237 |
| Omnidata-v1 [17] | 75.1M/123.1M | 0.0501 | 0.0750 | 0.1280 |
| X-task consistency [68] | 75.5M | 0.0502 | 0.0736 | 0.1390 |
| X-task baseline [68] | 75.5M | 0.0511 | 0.0763 | 0.1388 |
| StyleGAN | 24.7M | 0.0834 | 0.1216 | 0.1019 |

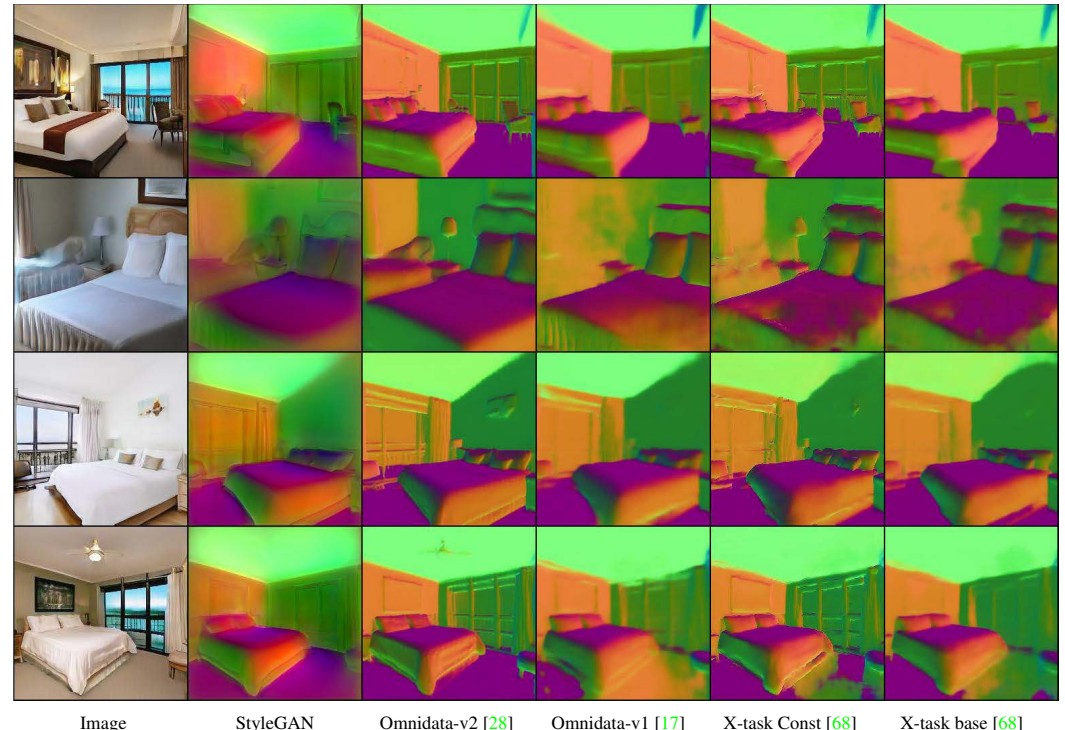

| Image | StyleGAN | Omnidata-v2 [28] | Omnidata-v1 [17] | X-task Const [68] | X-task base [68] |

Figure 3: **Normal generation.** StyleGAN generated normals are not as sharp as supervised SOTA methods but produce similar and accurate representations when compared to other methods.

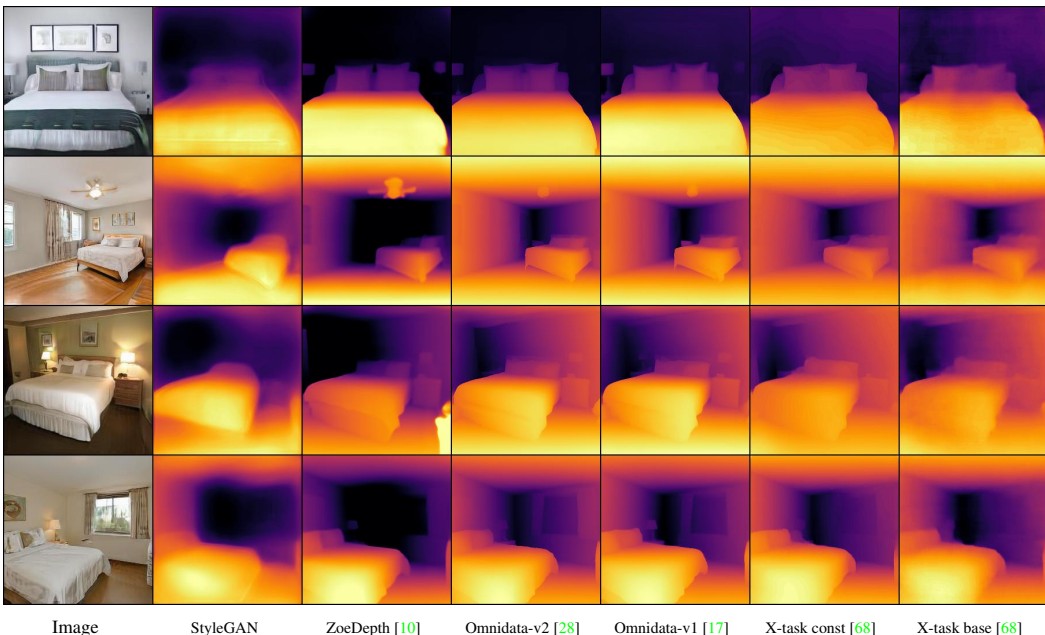

| Image | StyleGAN | ZoeDepth [10] | Omnidata-v2 [28] | Omnidata-v1 [17] | X-task const [68] | X-task base [68] |

Figure 4: **Depth estimation.** While fine details may not be as clearly estimated as top methods like ZoeDepth, the overall structure produced by StyleGAN is consistent in quality with recent models.

Table 2: **Quantitative Comparison of Segmentation:** Accuracy (Acc) and mean intersection over union (mIOU) are reported when compared to EVA-2 [18] as ground truth.

|  | bed | | pillow | | lamp | | window | | painting | | Mean | |
|---|---|---|---|---|---|---|---|---|---|---|---|---|
|  | Acc ↑ | mIoU ↑ | Acc ↑ | mIoU ↑ | Acc ↑ | mIoU ↑ | Acc ↑ | mIoU ↑ | Acc ↑ | mIoU ↑ | Acc ↑ | mIoU ↑ |
| DPT [44] | 97.1 | 93.2 | 98.9 | 82.5 | 99.6 | 79.3 | 99.0 | 90.3 | 99.6 | 92.9 | 98.8 | 87.7 |
| StyleGAN | 95.8 | 90.4 | 97.9 | 76.6 | 99.3 | 71.9 | 98.6 | 87.1 | 99.0 | 84.0 | 98.1 | 82.0 |

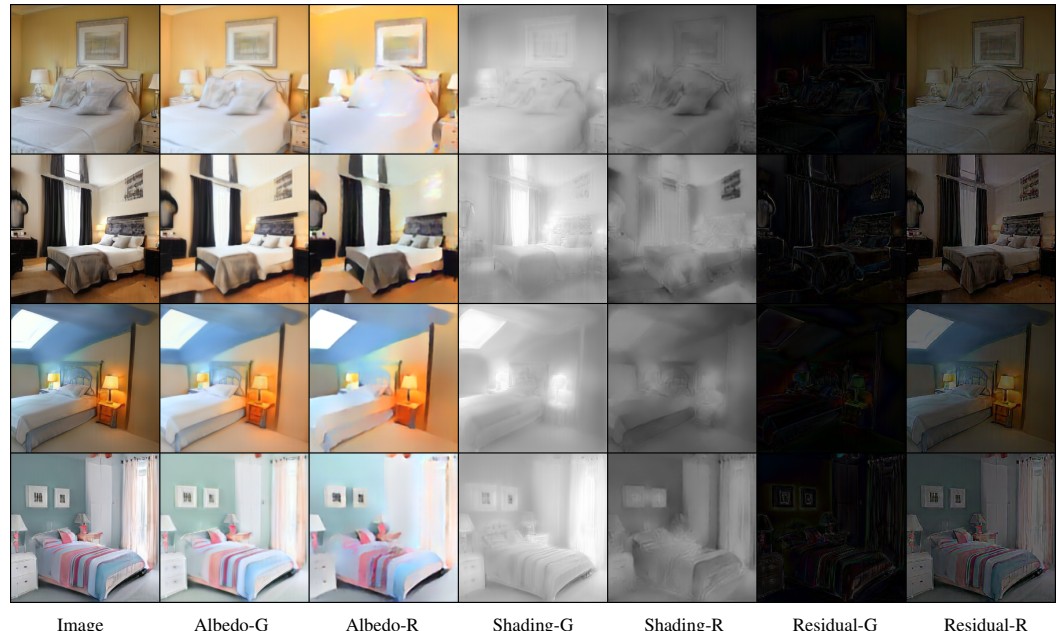

| Image | Albedo-G | Albedo-R | Shading-G | Shading-R | Residual-G | Residual-R |
|---|---|---|---|---|---|---|

Figure 5: **Albedo-Shading Recovery with StyleGAN**. We denote results from StyleGAN as -G and from a SOTA regression model [20] as -R. Absolute value of image residuals ($\mathcal{I} - \mathcal{A} * \mathcal{S}$) appears in the last two columns. While our searches for albedo and shading directions were independent, StyleGAN appears to "know" these two intrinsics should multiply to yield the image.

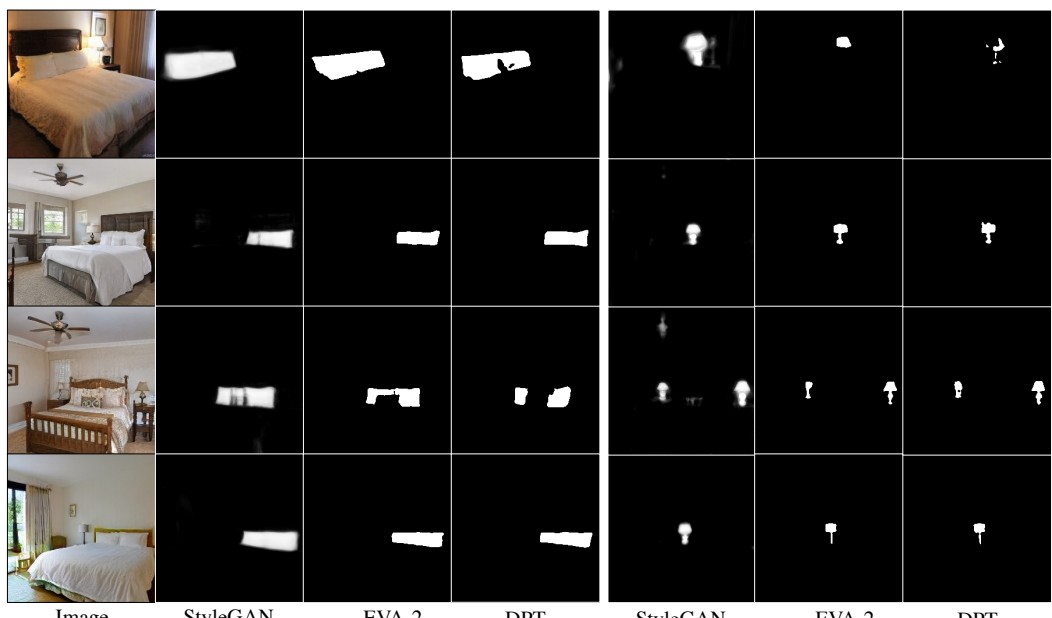

| Image | StyleGAN | EVA-2 | DPT | StyleGAN | EVA-2 | DPT |
|---|---|---|---|---|---|---|

Figure 6: **Segmentation Estimation**.Generated images with segmentation produced by the following methods: StyleGAN, EVA-2 [18], DPT [44] for `pillows` on the left and `lamps` on the right. Note that our quantitative comparison in Table 2 to a SOTA segmenter [18] likely *understates* how well StyleGAN can segment; for `lamps`, StyleGAN can find multiple lamps that the segmenter misses, likely because it put them there in the first place.

As shown in Figure 5, StyleGAN's generated albedo and shading results display significantly smaller residuals when contrasted with those from the SOTA regression-based image decomposition model.

**Segmentation:** We employ a supervised SOTA model, EVA-2 [18] and the top-performing segmentation method on the ADE20k benchmark [73] to guide our search for segmentation latents.

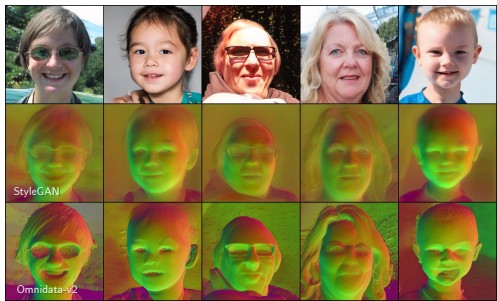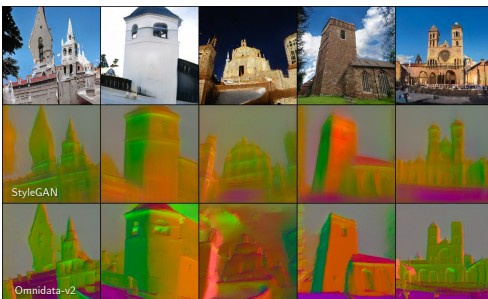

Figure 7: Examples of StyleGAN-generated normals for a StyleGAN model that was trained on the FFHQ dataset (left) and LSUN church (right). Top rows show randomly generated images, middle rows display StyleGAN-generated normals with fixed offset to $w^+$ codes of top rows and last rows provide Omnidata-v2 Surface Normals for comparison.

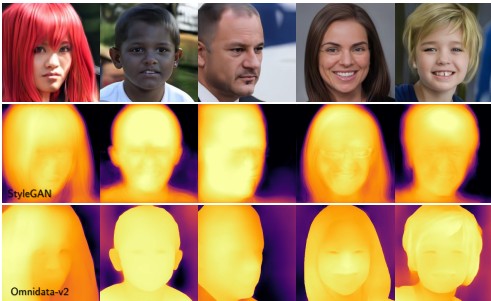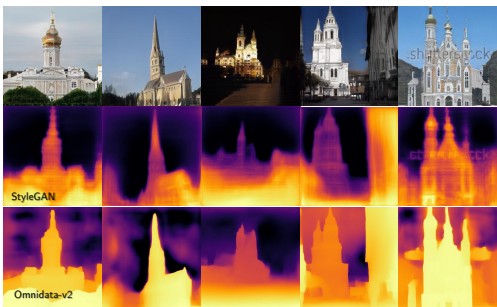

Figure 8: Examples of StyleGAN-generated depth for a StyleGAN model that was trained on the FFHQ dataset (left) and LSUN church (right). Top rows show randomly generated images, middle rows display StyleGAN-generated depth with fixed offset to $w^+$ codes of top rows and last rows provide Omnidata-v2 depth predictions for comparison.

Given StyleGAN's restriction to synthesizing only 3-channel output, we focus our search for directions that segment individual objects. Moreover, we use a weighted regression loss to address sparsity in features like lamps and pillows, which occupy only a minor portion of the image.

A quantitative comparison of our StyleGAN-generated segmentation for five different object categories is shown in Table 2, with qualitative comparisons in Figure 6. The accuracy of StyleGAN-generated segmentation is quantitatively comparable to a large vision-transformer-based baseline DPT [44]. Furthermore, qualitative results show that the segmentation of lamps and pillows generated by StyleGAN is more complete and slightly better to those from SOTA methods.

**Control: StyleGAN cannot do non-intrinsic tasks.** While our search is over a relatively small parameter space, it is conceivable that StyleGAN is a generic image processing engine. We check that there are tasks StyleGAN will *not* do by searching for an offset that swaps the left and right halves of the image (Figure 9).

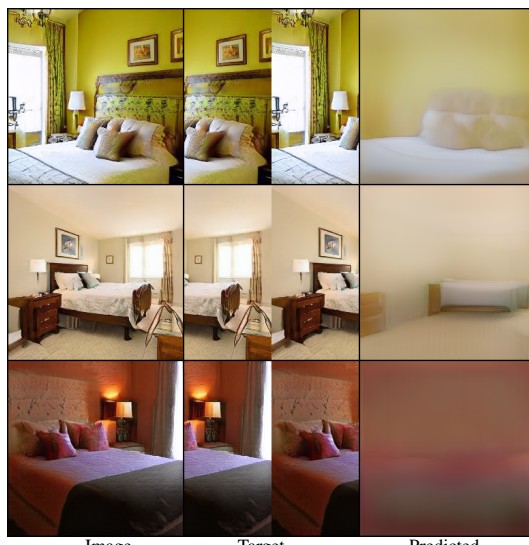

Image    Target    Predicted

Figure 9: StyleGAN is not a generic image processing machine; for ex., it cannot swap left and right halves of an image. This supports the conclusion that StyleGAN trades in "characteristics ... of the surface element visible at each point of the image" or intrinsic images.

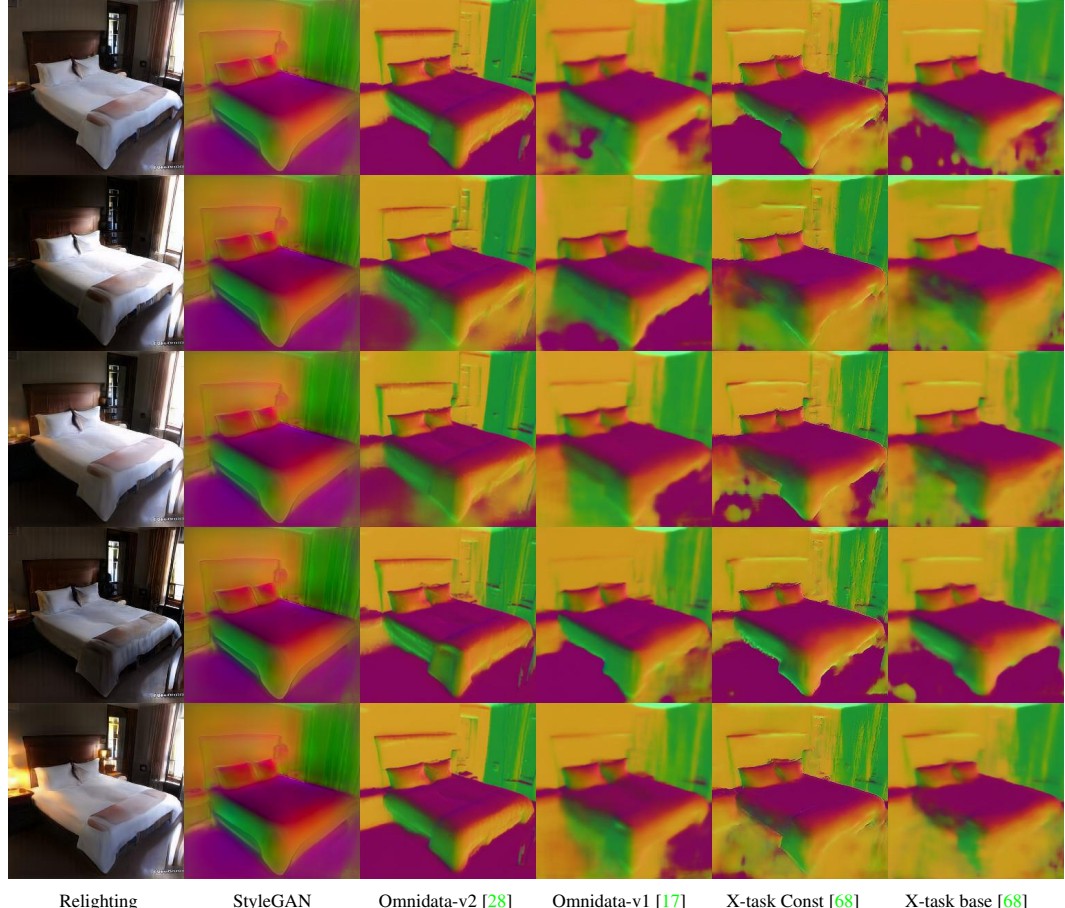

| Relighting | StyleGAN | Omnidata-v2 [28] | Omnidata-v1 [17] | X-task Const [68] | X-task base [68] |

Figure 10: **Robustness against lighting changes.** Normals should be invariant to lighting changes, yet surprisingly, the top-performing regression methods, even those trained with labeled data, with lighting augmentations and 3D corruptions [28], demonstrate sensitivity to lighting alterations. Intriguingly, StyleGAN-generated normals prove to be robust against lighting alterations.

## 6    Robustness of StyleGAN Intrinsics

We investigate the sensitivity of predicted normals, depth, segmentation, and albedo to alterations in lighting. For this purpose, we utilize StyLitGAN [11], which generates lighting variations of the same scene by introducing latent perturbations to the style codes. This allows us to add $\mathbf{d}(relighting)$ to $\mathbf{w}^+$ to create different lighting conditions of the same scene, and then add $\mathbf{d}(\mathbf{c})$ to predict the corresponding scene property maps. Ideally, the predicted normals, depth, segmentation, and albedo should remain invariant to lighting changes.

However, we observe that the current state-of-the-art regression methods exhibit sensitivity to such lighting changes. Interestingly, the intrinsic predictions generated by StyleGAN demonstrate robustness against these lighting variations, significantly surpassing their state-of-the-art counterparts in terms of performance. Note that comparison methods use much larger models and extensive data augmentations. A qualitative comparison is in Figure 10 and a quantitative analysis for variation in surface normals is in Figure 11. A similar trend is observed for other intrinsics.

## 7    Discussion

Our study, while exploratory, has broad implications. We discuss a few of them below.

**Understanding Generative Models.** We highlight important properties and capabilities in generative models when not trained with explicit supervision. This insight could potentially lead to the development of more efficient and effective training methods for generative models.

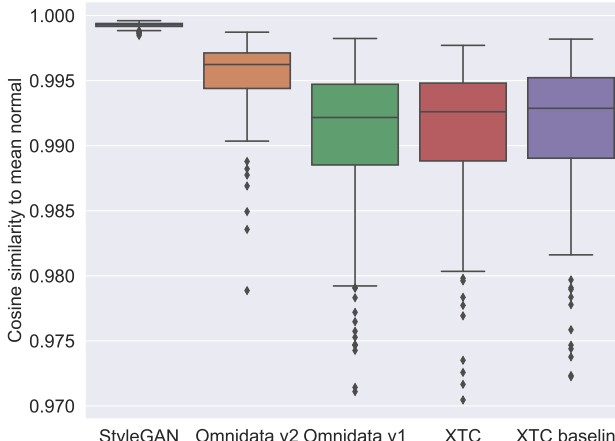

Figure 11: **Quantitative evaluation of normal variations following relighting.** Normals are calculated under 16 distinct lighting conditions from [11] for 214 test scenes for each normal prediction methods. The inner product between normals under each condition and the overall mean is computed. Ideally, this should be 1, indicating normals' consistency. The boxplots illustrate these values, with regression methods showing an average change of 8 degrees, with outliers up to 14 degrees. High similarity scores for StyleGAN indicate its normals' robustness to lighting changes.

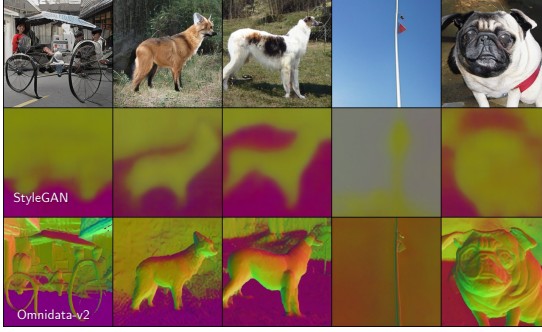

Figure 12: This figure shows the failure in generation of surface normal directions using StyleGAN-XL on ImageNet with 10,000 unique generated images. The images in the middle row resemble the surface normals in the bottom row but miss details that [28] predicted. The differences in architecture between StyleGAN-XL and the more appropriate StyleGAN-2 make it difficult to leverage this model. Either an alternate strategy is required or the latent space in this model is not easy to manipulate.

**Intrinsic Images from other Generative Models.** We have demonstrated that StyleGAN can easily extract representations of important intrinsic images. This makes our work the first to achieve this, and that too by simply adding an offset to a pretrained generative model. Even if the current method has a performance gap compared to SOTA monocular predictors, this understanding that generative models can capture such intrinsic properties provides a new avenue for research. Our findings raise compelling questions for further investigation. We suspect that this property may also be true for other generative models. Is this the case, or is it a result of specific design choices in StyleGAN? We think other generative models "know" intrinsic images, too, but don't currently know how to elicit that knowledge.

**Robust Intrinsic Images.** One of the potential advantages of our approach is deriving intrinsic images that are inherently more robust, especially against variations like lighting changes (Figure 10). One might use this to bootstrap a better StyleGAN predictor from a non-robust, but SOTA, predictor.

**Guidance from SOTA Monocular Predictor.** Currently, we rely on monocular predictors for offset guidance. However, we use limited guidance, consisting of just 2000 examples. We anticipate that future models may not require such guidance.

**GAN Inversion.** To build a practical intrinsic image predictor from a StyleGAN, one needs an inverter that is (a) accurate and (b) preserves the parametrization of the image space by latents – can this be built?

**What Makes Good Image Representation?** Finally, our findings provide insights into what makes a "good" image representation. Our discovery of fundamental image intrinsics such as depth, albedo, and normals in the latent space of a StyleGAN suggests that meaningful image representations are those that can capture these intrinsic properties, as highlighted in Barrow and Tenenbaum's work [6]. StyleGAN clearly 'knows' the intrinsics that are well-understood in the vision and graphics community. Are there other intrinsics that it 'knows' but that we have overlooked? The materials literature, particularly studies like Motoyoshi et al. [38] and Sharan et al. [51], suggest that many other types of intrinsic images may exist. Therefore, our search method could potentially uncover unknown intrinsic images.

## Acknowledgment

This material is based upon work supported by the National Science Foundation under Grant No. 2106825 and by gifts from Amazon and Boeing.

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

## Appendix

In Figure 13, we demonstrate the segmentation quality with increase in number of labeled examples. We also provide additional qualitative figures for intrinsic image predictions from StyleGAN – normals in Figure 14, depth in Figure 15, albedo-shading decomposition in Figure 16, and segmentation of lamps and pillows in Figure 17, segmentation of windows and paintings in Figure 18 and segmentation of beds in Figure 19. In our experiments, we generated three-channel output for each depth, shading, and segmentation from StyleGAN. We took the mean for each of them to get the final single-channel estimate. We also provide additional examples of robustness against lighting changes for the segmentation task in Figure 20.

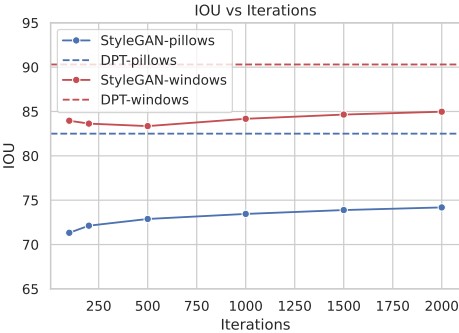

Figure 13: Segmentation Quality vs. Labeled Dataset Size: The graph illustrates how segmentation quality for 'pillows' and 'windows' classes improves with increasing labeled images, plateauing around 2,000 images. Note that all images are seen only once. While the potential exists for further refinement, the core insight is StyleGAN's inherent capability to generate these segmentations without explicit training.

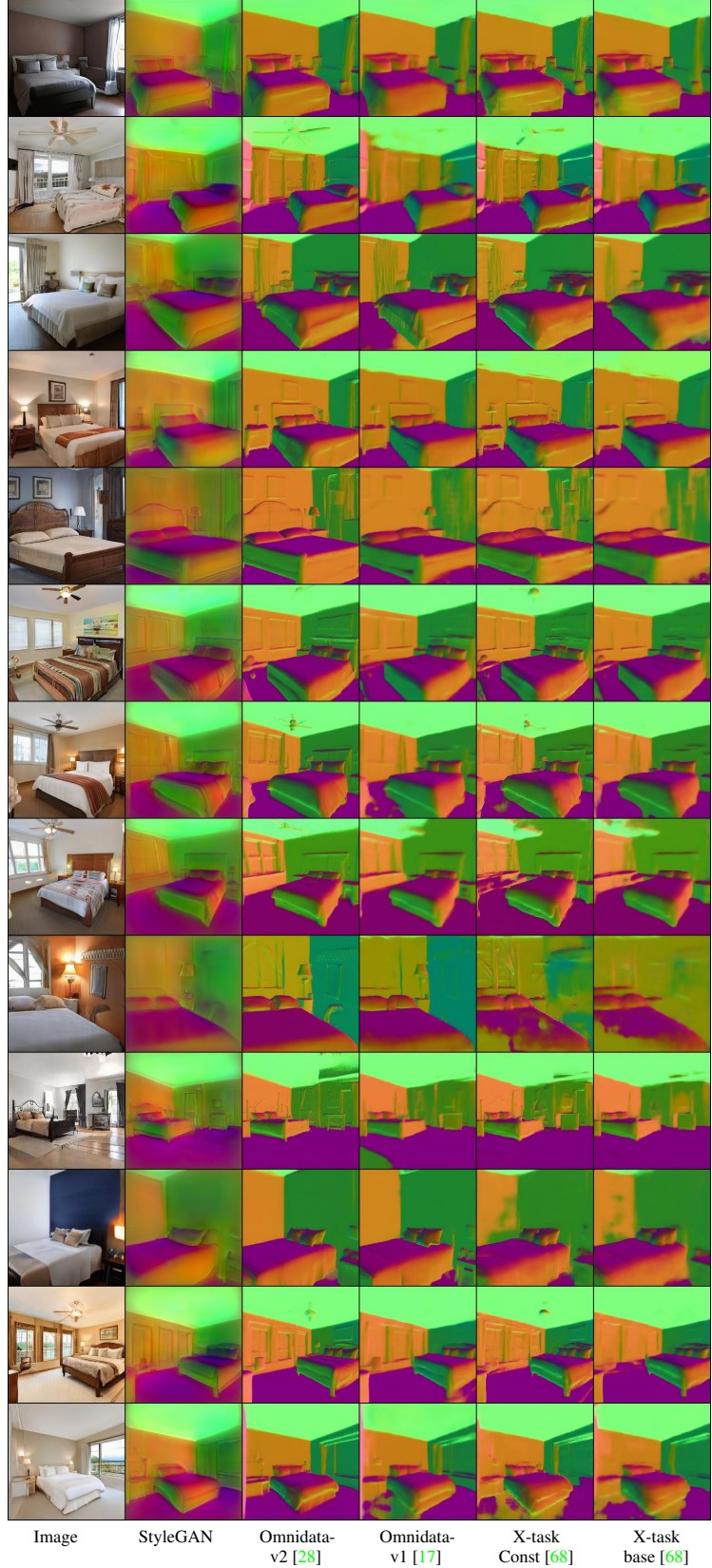

| Image | StyleGAN | Omnidata-v2 [28] | Omnidata-v1 [17] | X-task Const [68] | X-task base [68] |

Figure 14: **Additional Normal generation.**

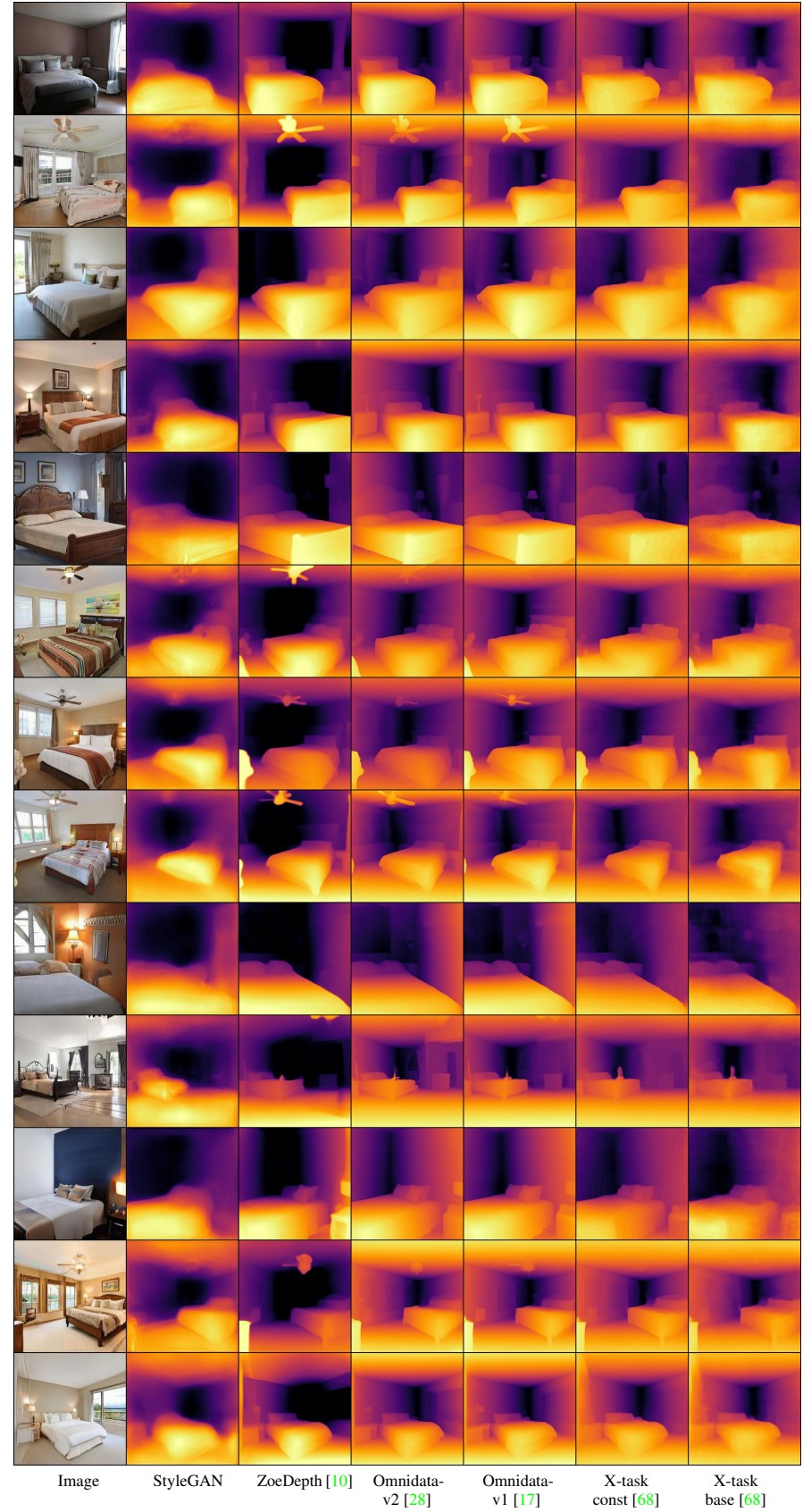

| Image | StyleGAN | ZoeDepth [10] | Omnidata-v2 [28] | Omnidata-v1 [17] | X-task const [68] | X-task base [68] |

Figure 15: **Additional Depth Estimation Comparison.**

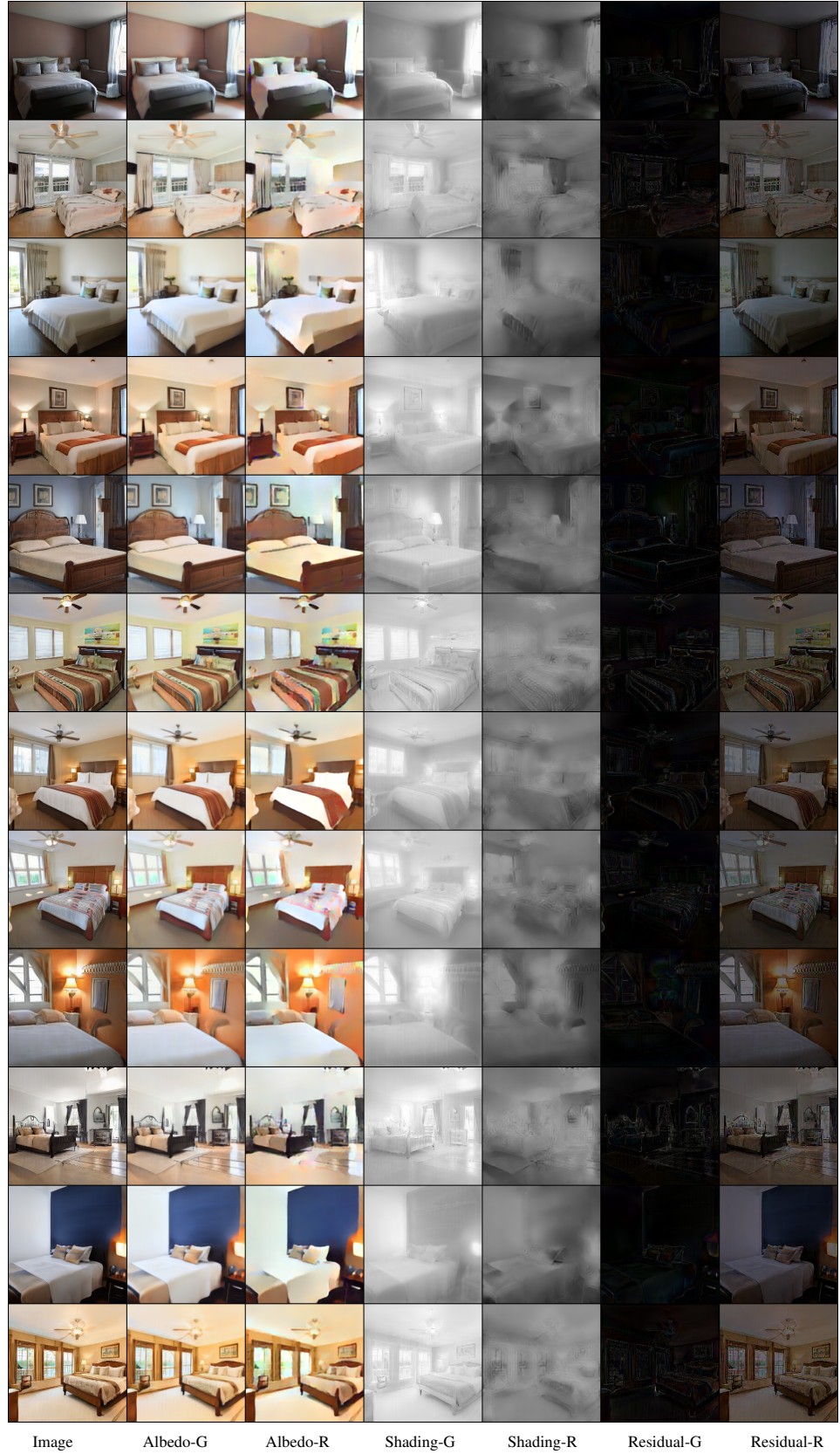

Image  Albedo-G  Albedo-R  Shading-G  Shading-R  Residual-G  Residual-R

Figure 16: **Additional Results for Albedo-Shading Recovery with StyleGAN**.

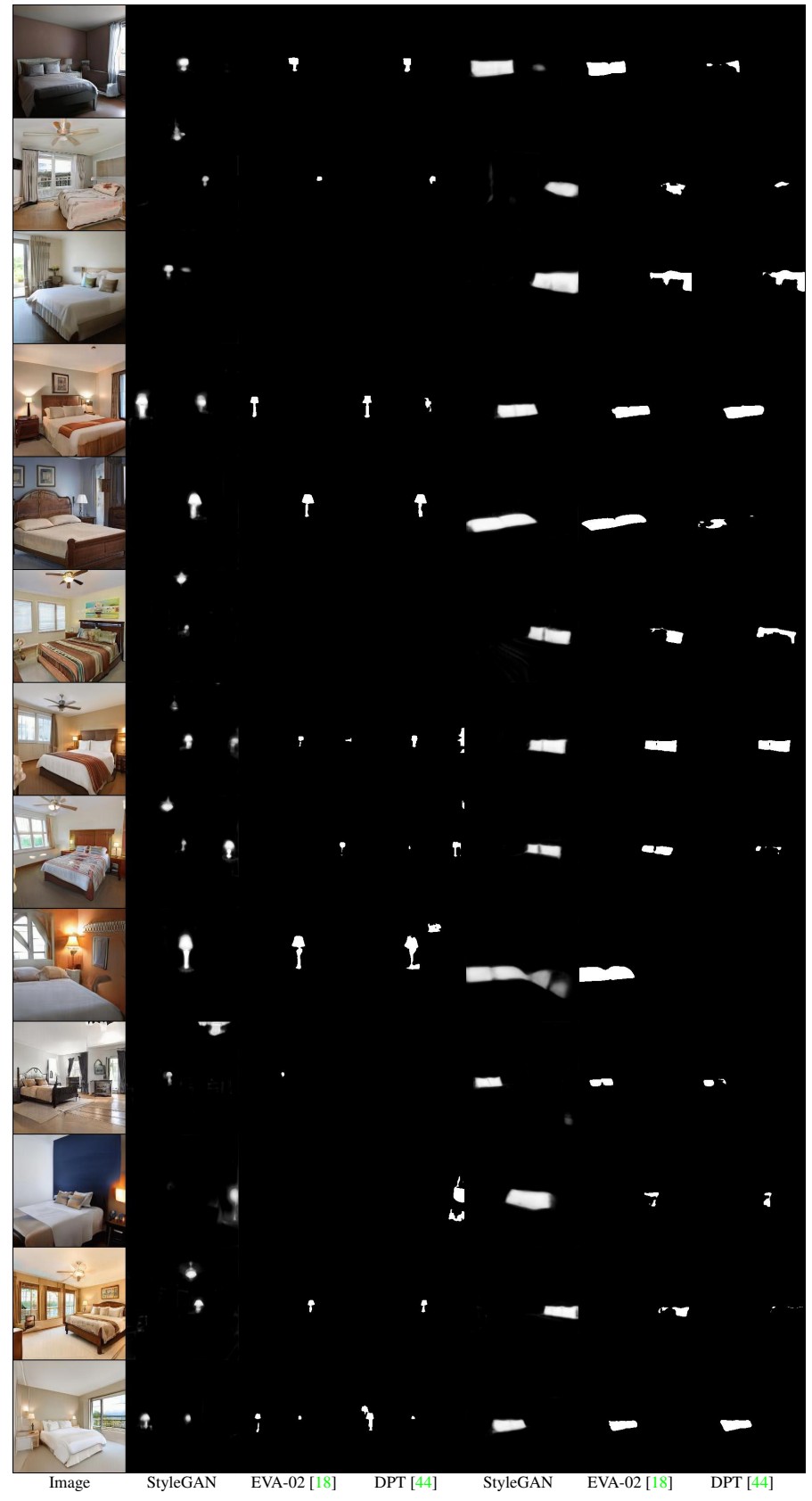

| Image | StyleGAN | EVA-02 [18] | DPT [44] | StyleGAN | EVA-02 [18] | DPT [44] |

Figure 17: **Further segmentation of lamps on the left and pillows on the right.**

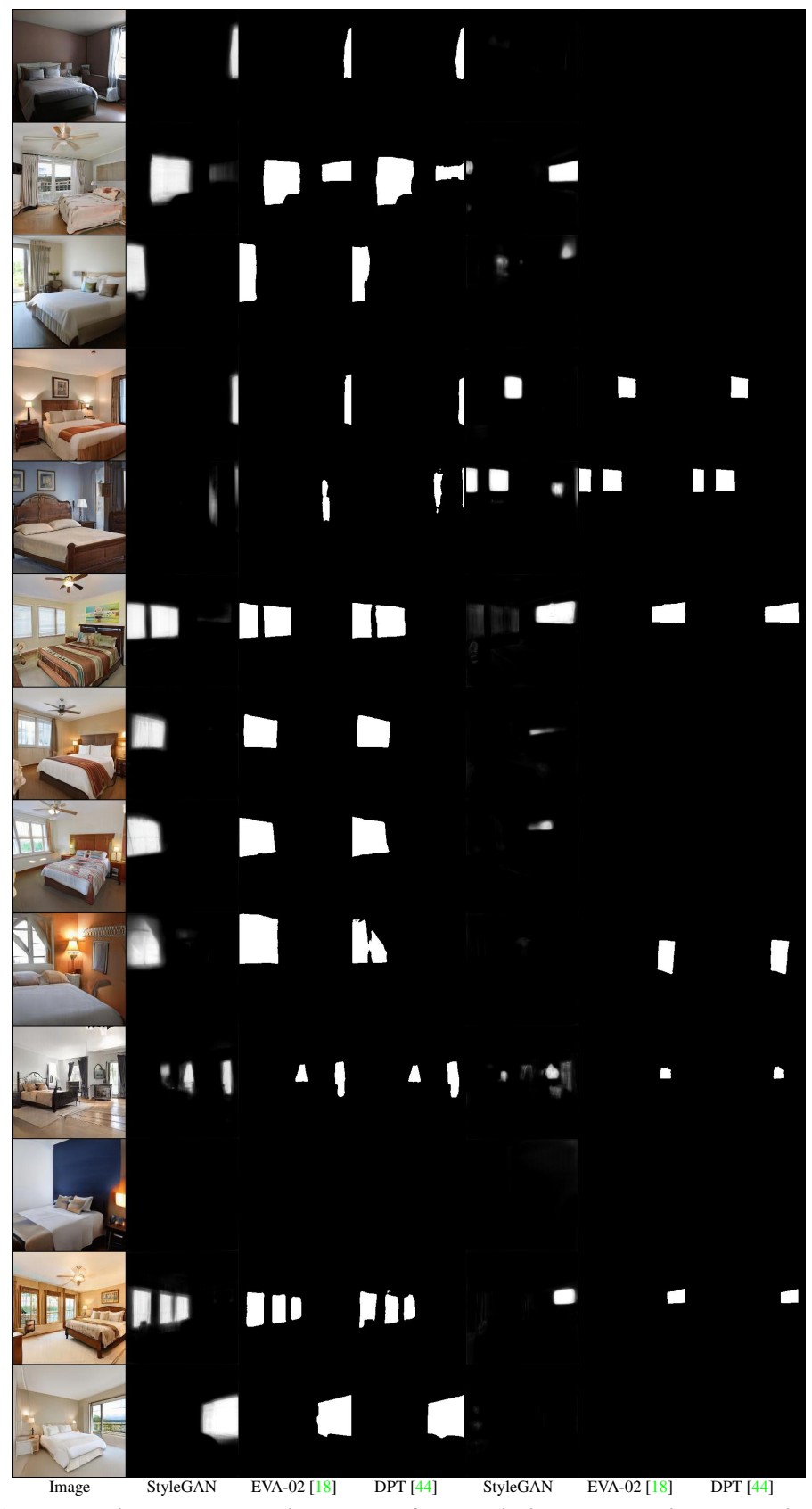

Image      StyleGAN    EVA-02 [18]    DPT [44]    StyleGAN    EVA-02 [18]    DPT [44]

Figure 18: **Window segmentation on the left and painting segmentation on the right.**

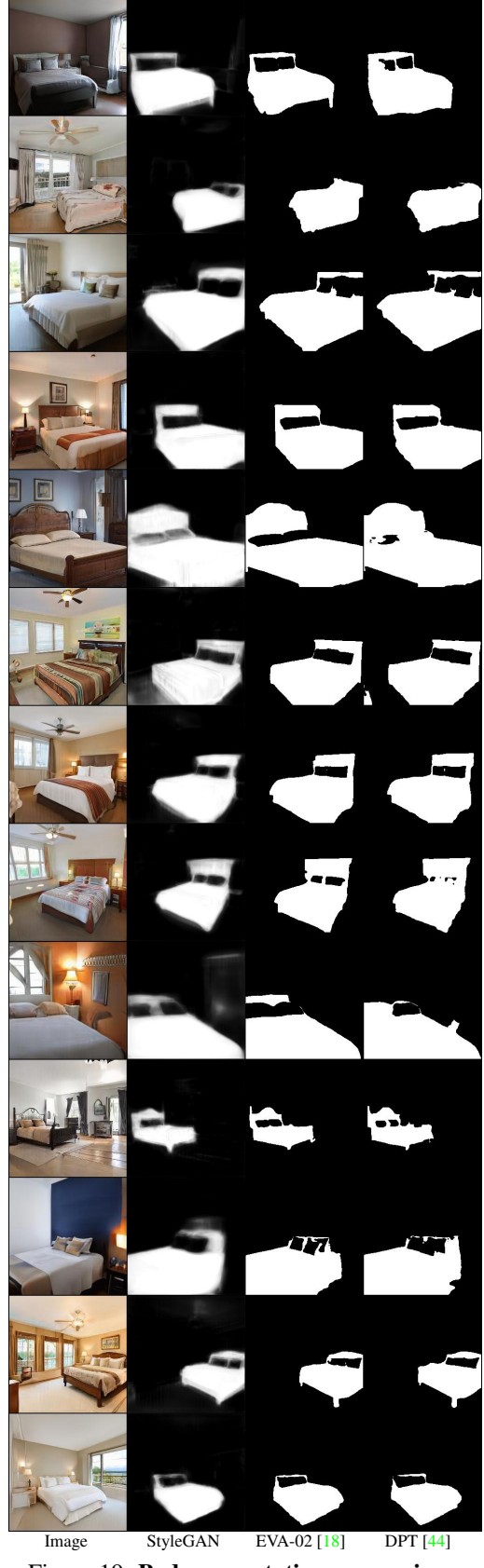

Image   StyleGAN   EVA-02 [18]   DPT [44]

Figure 19: **Bed segmentation comparison.**

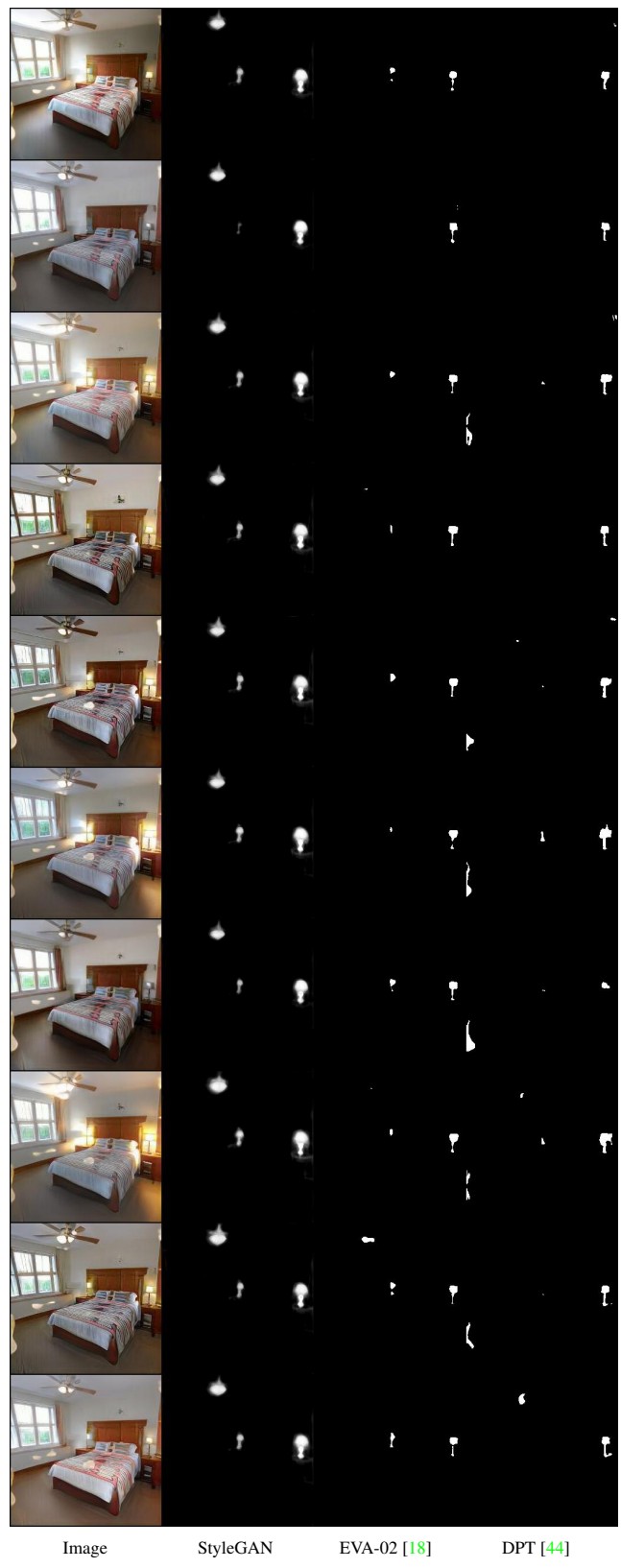

Image      StyleGAN      EVA-02 [18]      DPT [44]

Figure 20: **Additional examples for robustness against lighting for segmentation.**

