## StyleGAN knows Normal, Depth, Albedo and More – Supplement

We provide additional qualitative figures for intrinsic image predictions from StyleGAN – normals in Figure 10, depth in Figure 11, albedo-shading decomposition in Figure 12, and segmentation of lamps and pillows in Figure 13, segmentation of windows and paintings in Figure 14 and segmentation of beds in Figure 15. In our experiments, we generated three-channel output for each depth, shading, and segmentation from StyleGAN. We took the mean for each of them to get the final single-channel estimate. We also provide additional examples of robustness against lighting changes for the segmentation task in Figure 16.

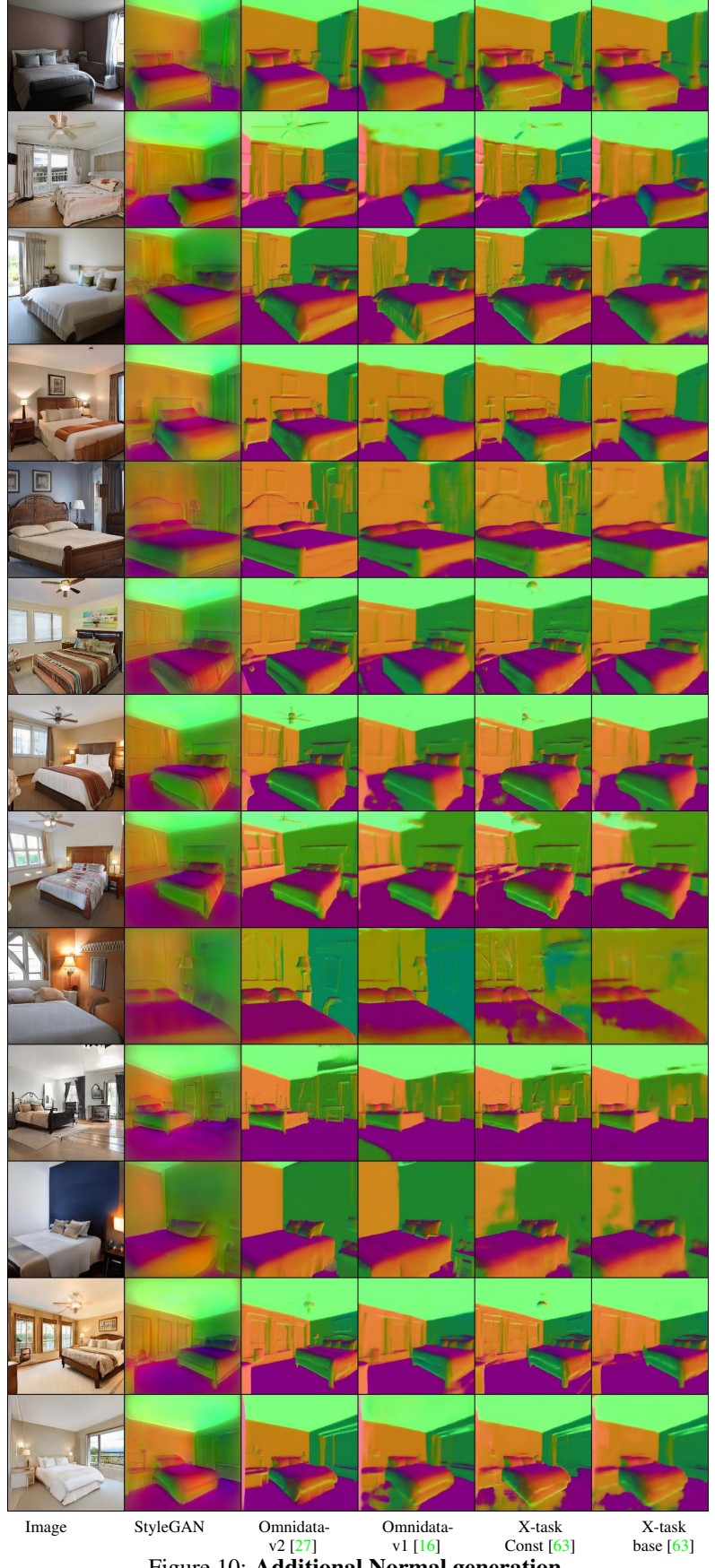

| Image | StyleGAN | Omnidata-v2 [27] | Omnidata-v1 [16] | X-task Const [63] | X-task base [63] |

Figure 10: **Additional Normal generation.**

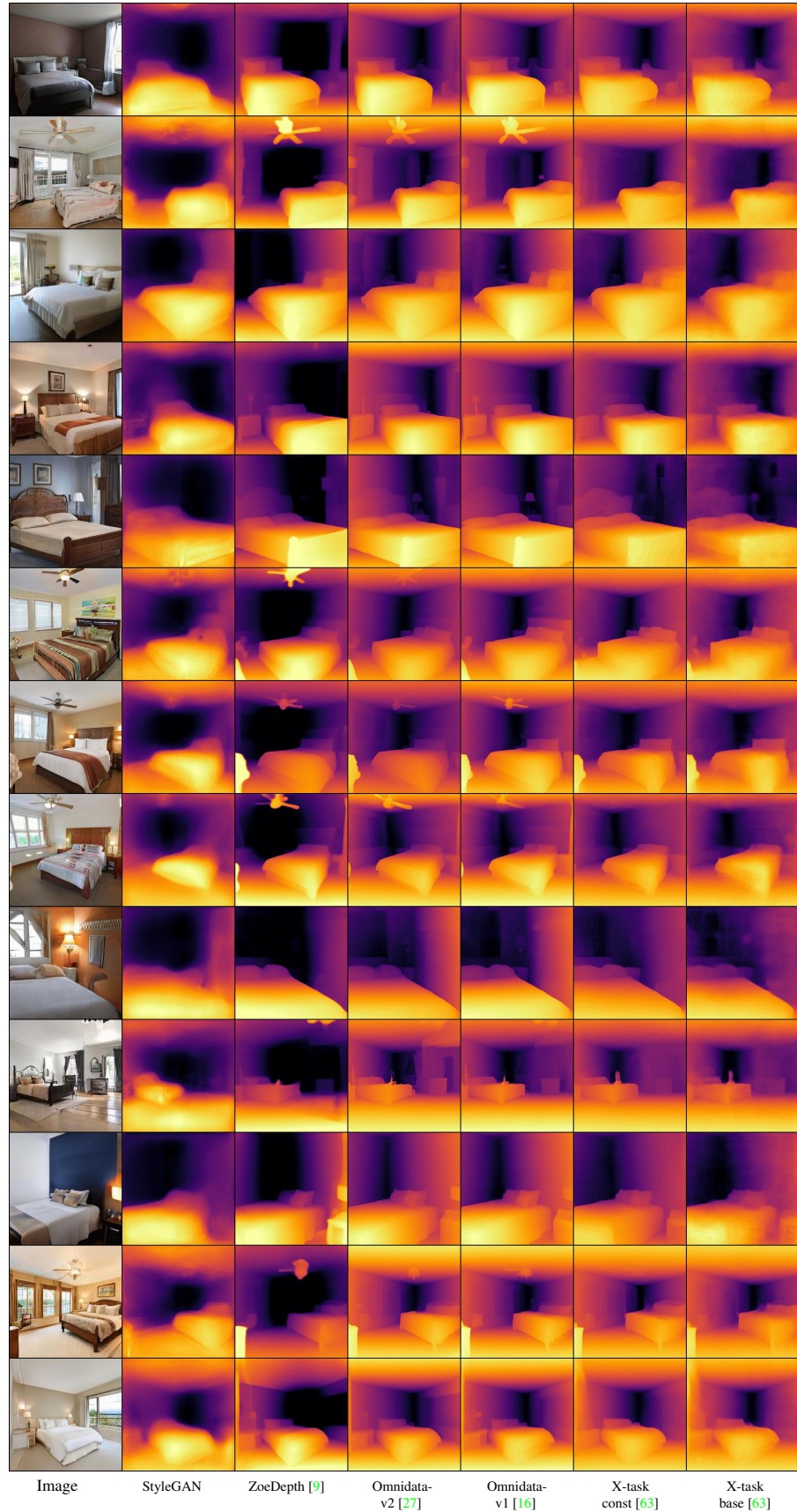

| Image | StyleGAN | ZoeDepth [9] | Omnidata-v2 [27] | Omnidata-v1 [16] | X-task const [63] | X-task base [63] |

Figure 11: **Additional Depth Estimation Comparison.**

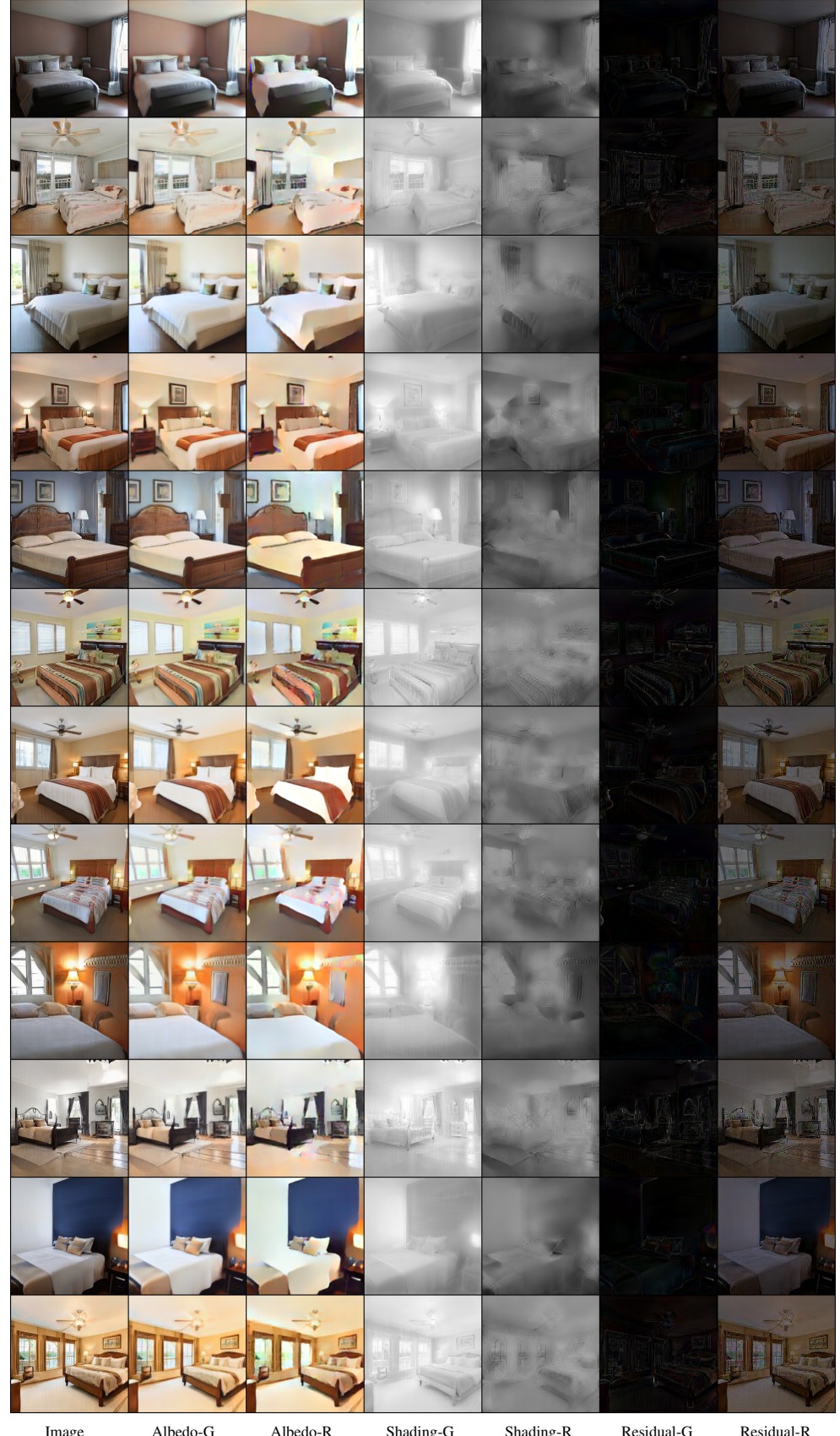

| Image | Albedo-G | Albedo-R | Shading-G | Shading-R | Residual-G | Residual-R |

Figure 12: **Additional Results for Albedo-Shading Recovery with StyleGAN**.

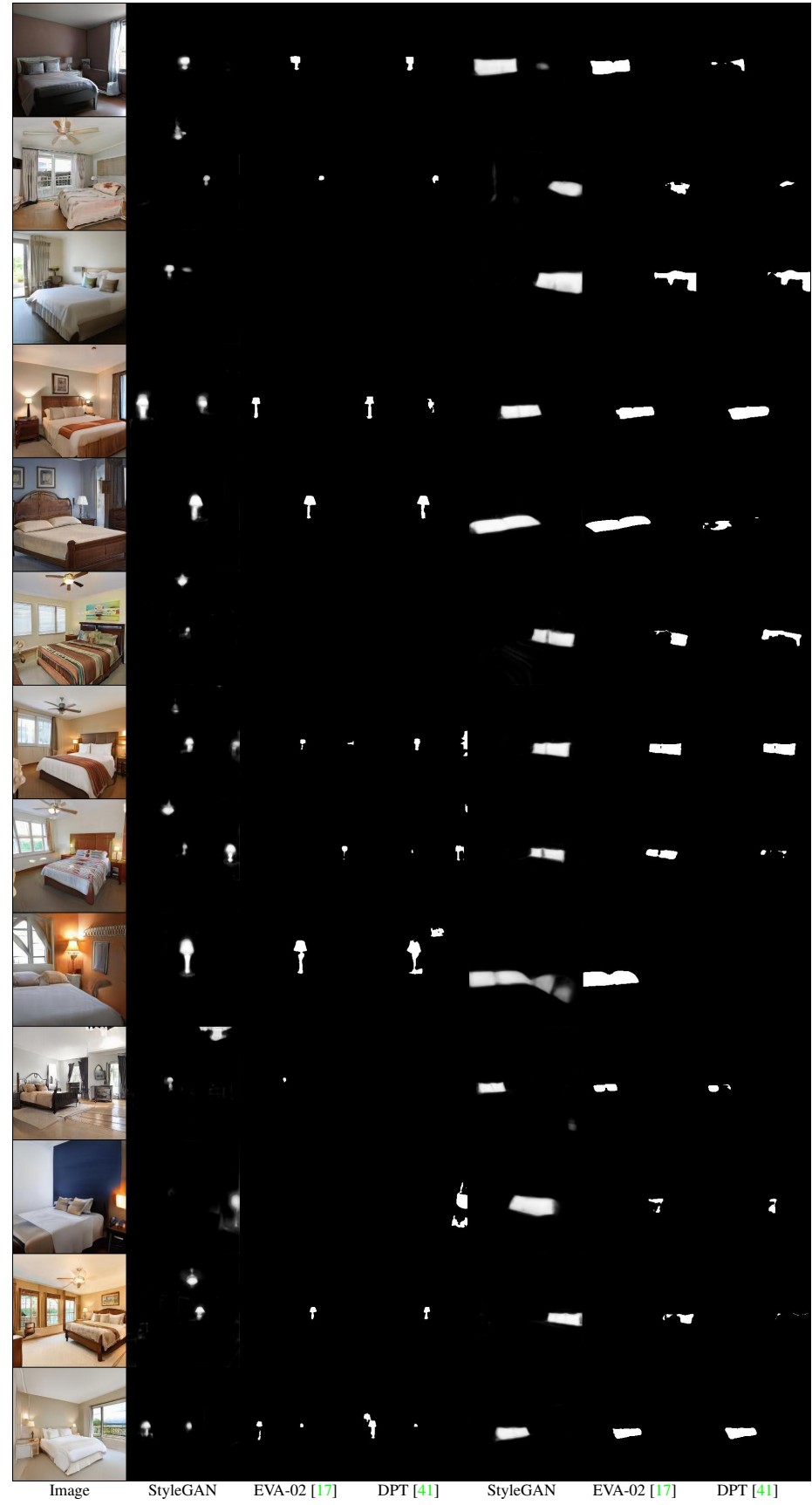

Image      StyleGAN      EVA-02 [17]      DPT [41]      StyleGAN      EVA-02 [17]      DPT [41]

Figure 13: **Further segmentation of lamps on the left and pillows on the right.**

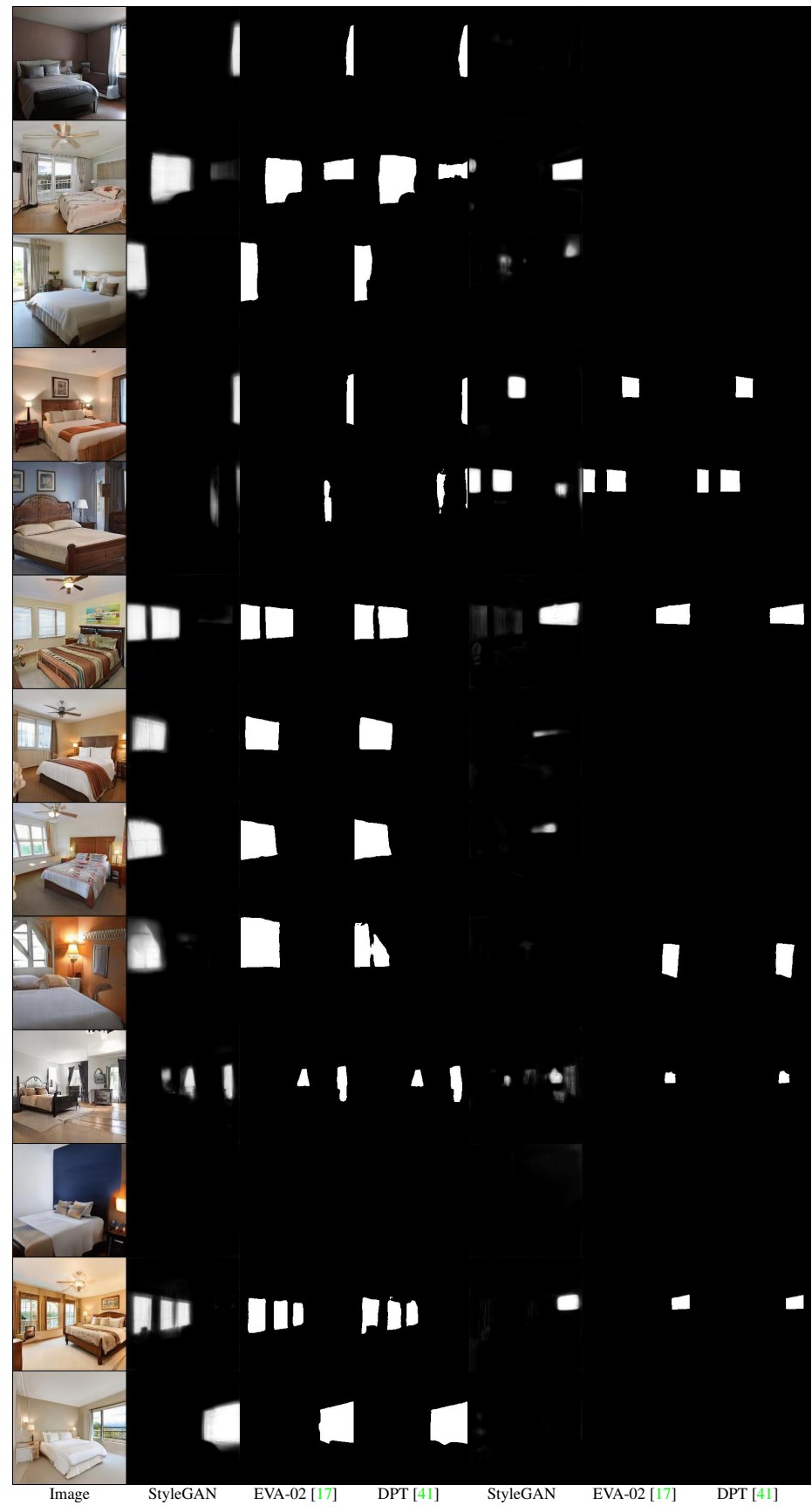

| Image | StyleGAN | EVA-02 [17] | DPT [41] | StyleGAN | EVA-02 [17] | DPT [41] |

Figure 14: **Window segmentation on the left and painting segmentation on the right.**

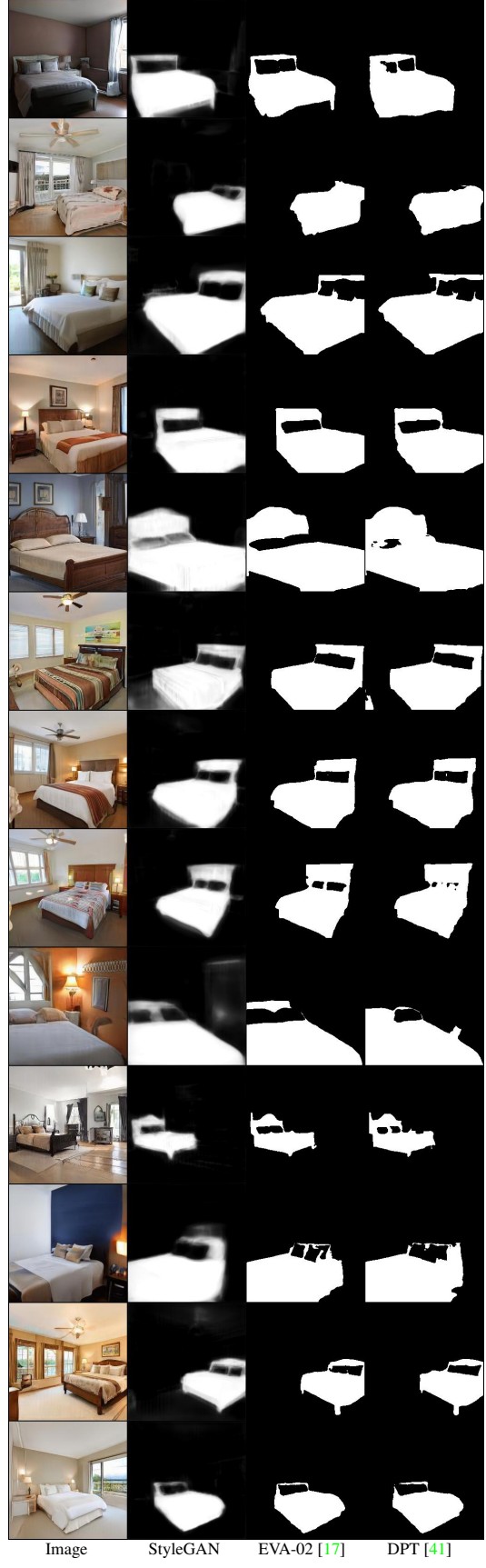

Image StyleGAN EVA-02 [17] DPT [41]

Figure 15: **Bed segmentation comparison.**

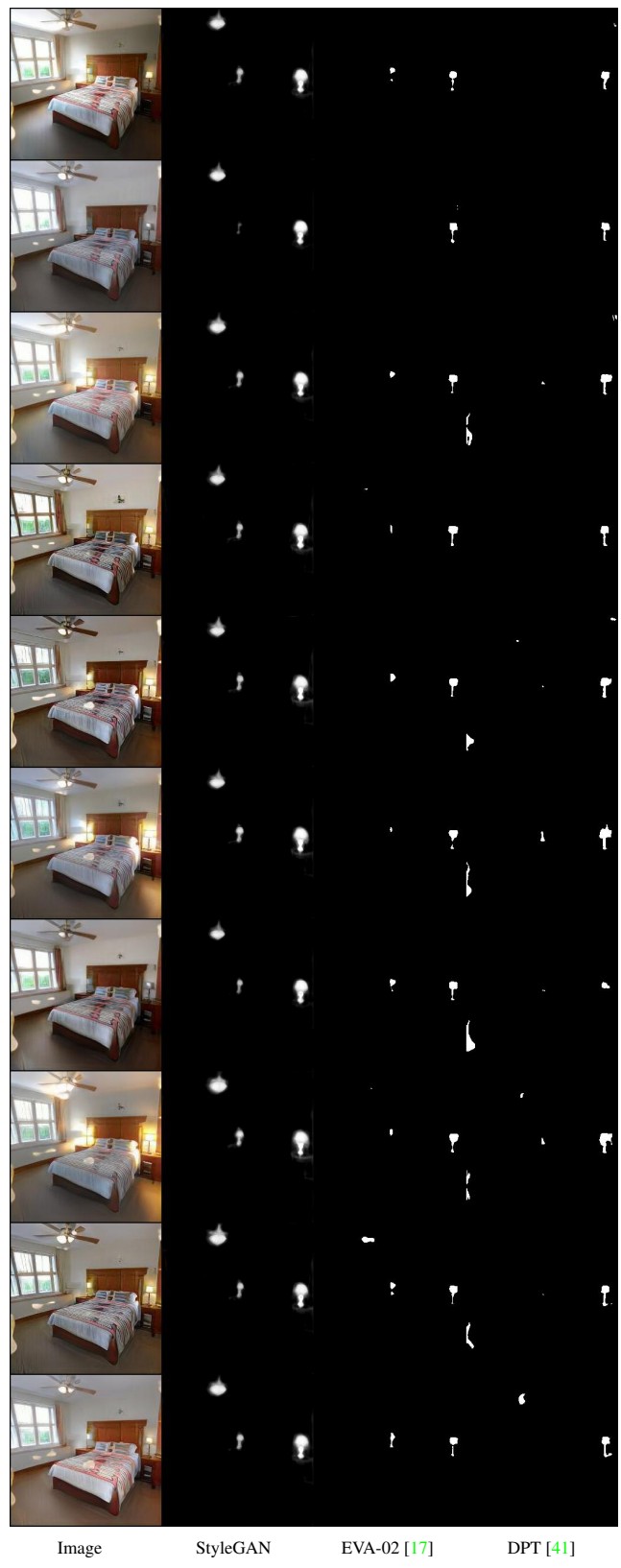

Image        StyleGAN        EVA-02 [17]        DPT [41]

Figure 16: **Additional examples for robustness against lighting for segmentation.**