# OpenReview forum: "StyleGAN knows Normal, Depth, Albedo, and More"
_NeurIPS.cc/2023/Conference — NeurIPS 2023 poster_

### Official Review · Reviewer_qbrH · 2023-06-19

**Soundness:** 3 good
**Presentation:** 2 fair
**Contribution:** 3 good
**Rating:** 5
**Confidence:** 4

**Summary:**

This paper proposes a novel method for obtaining "intrinsic images" for an RGB image, where in this context intrinsic images mean mainly depth, normal, segmentation, albedo, and shading maps. To this end, the authors investigate how a pre-trained StyleGAN generator model can be used to output the respective intrinsic image instead of an RGB image by only adding a constant in style weight space. More specifically, the proposed method optimizes an offset vector in the StyleGAN "w+" style space for a given editing constraint. The offset vectors are optimised via gradient descent where a comparably small dataset of generated images together with respective intrinsic images, predicted by SOTA models, is used as guidance information. The authors compare the proposed method to SOTA methods and find that it performs comparably.

**Strengths:**

- The proposed approach is an interesting and valuable method for investigating what information is already contained in generative models such as StyleGAN. The finding that adding an offset vector in style weight space to output one of the other data modalities instead of RGB is surprising and very interesting. It highlights how much "semantic and 3D scene understanding" can be found in 2D generative models.
- The authors investigate the performance of the same proposed method with multiple different data modalities highlighting the generality of the approach.
- I appreciate that the authors also show an example decomposition that clearly does not work with the given method (L. 178). As only a style weight space vector is added, non-pixel-based transformations are expected to not work well / fail.
- The authors compare against relevant baselines quantitatively and qualitatively and show helpful visual comparisons.

**Weaknesses:**

- Performance across different datasets / Dependence on small labeled dataset: The proposed method requires a sub-set of "labeled" generated image and intrinsic image pairs, where the latter is provided by a SOTA method. Have the authors investigate how well the method performs if this "fine-tuning" is done on a different dataset, or on only a sub-set of the covered "modes"? More general, investigating different factors of this training stage with the resulting end performance, e.g. in form of a graph showing number of training images against performances, would be very interesting.
- Labeled dataset: In similar spirit, how many labeled images (provided by the SOTA method) are used? (L. 124 - 126 is vague on this, and it would be beneficial to state clear numbers.)
- Evaluation: The authors argue that no ground truth is available, hence SOTA predictions are used as GT. But couldn’t the test set of these SOTA methods be used as GT? For the proposed method, the input image would be needed to be inverted in StyleGAN space, but this has been extensively done before in the literature. Then, the proposed method could directly be compared to the SOTA prediction model itself.
- Depth Comparison: It is unclear to me why the proposed method performs better in the L1 depth metric (Table 1) while it looks qualitatively worse than the other methods. Could the authors expand on this? I am not sure if this is caused by some scale/shift misalignments; as these are monocular predictors, the scale and shift is arbitrary, so this needs to be taken into account during evaluation.
- Caption Fig 6: “Note that our quantitative comparison in Table 2 to a SOTA segmenter [17] likely understates how well StyleGAN can segment;” -> Can the authors expand on this? This is not clear to me. The authors give the two-lamp example, however, here all methods were able to segment the two lamps.

**Questions:**

- How sensitive is the proposed system to the number of image "labeled" image pairs used for training?
- Could the GT image and data modality pairs of the test set of the SOTA predictors be used as GT for evaluation?
- Why is the proposed system better in the L1 depth metric while it looks worse qualitatively?
- Is the same offset style vector d(c) used across all images for the same intrinsic image modality?
- How many images of the "labeled" datasets are used for optimising the offset style vectors?

For more context on these questions, please see the "Weakness" paragraph.

The manuscript contains the following typos / other formatting errors:

Typos:

	Caption Fig 1: “StyleGAN has easily accessed and accurate representations of intrinsic images” is unclear and grammatically wrong. What does this mean?
	Caption Fig 1: ““knows” fine detail in normal” -> “knows” fine detail in normal maps
	L. 49: “Showing that intrinsic images so extracted” -> “Showing that these intrinsic images extracted from StyleGAN …
	L. 54: “and produce blurry outputs” -> and producing blurry outputs
	L. 91: “significant recent methods” -> Important recent methods
	L. 92: “Learned methods” -> Learning-based methods
	Caption Fig 2: “We demonstrate a StyleGAN that is trained to generate images encode accessible scene property maps” -> […] is trained to generate images encodes accessible scene property maps
	Caption Fig 2: “Our approach explores [….], generate” -> Our approach explores […], generates
	L. 123: “a select number” -> a selected number
	L. 198: “We have demonstrated that StyleGAN has easily extracted representations” -> We have demonstrated […] has easily extractable representations
	L. 200: “be true of other generative models” -> be true for other generative models

Other:

	Figure 1 and Figure 2 are not referenced in the text. Please add a reference to the main text.

**Limitations:**

- The authors show and discuss and example data modality / transformation that is cannot be tackled with the proposed approach due to the pixel-aligned features (L. 178).
- The authors investigate the performance of the proposed system for different lighting conditions of the images (L. 184).
- In the discussion (Sec. 7), the authors discuss unclear points that would require additional investigation. This includes to what extent these findings translate to other generative models and if more intrinsic image modalities next to the studies ones are contained in StyleGAN's weight space.
- The authors do not discuss potential negative societal impact.

---

> ### Author Rebuttal · Authors · 2023-08-10
>
> Thank you, Reviewer qbrH, for your insightful feedback and acknowledgment of the strengths in our work. We are particularly heartened by your recognition of our proposed approach as an “interesting and valuable method” for investigating the information contained within generative models like StyleGAN. Your emphasis on our “surprising and very interesting findings” and the highlighting of the “generality of our method” reinforces the core strengths of our research. We also appreciate your keen observation of our example decomposition and the expectation regarding non-pixel-based transformations. Lastly, your positive note on our comparison against relevant baselines and the inclusion of helpful visual comparisons is greatly valued. We will now address the points raised in your review.
>
> **Performance Across Datasets & Dependence on Labeled Data:** We appreciate the reviewer's emphasis on the generalizability of our method. We now have expanded our evaluations to include additional datasets beyond the LSUN Bedroom. These new evaluations on the FFHQ and LSUN bedroom model firm the consistency of our method's performance across diverse datasets, reinforcing its robustness. Please see our rebuttal's PDF files for results on these datasets.
>
> Furthermore, to address concerns regarding the dependence on the size of the labeled dataset, we have included a new plot illustrating the relationship between the number of labeled images and the quality of the generated segmentations for pillows and windows class. This graph spans a range from 100 to 2,000 labeled images. Our findings indicate an increase in performance as we increase the labeled dataset and start saturating around the 2000 images mark. The performance does vary, showing improvements with more labeled pairs. With more labeled examples and a better search procedure, we may get better intrinsic images from StyleGAN. However, the goal of the paper is not aimed to outperform SOTA methods but to demonstrate intrinsic images are baked into the latent space of StyleGAN without being explicitly trained to do so.
>
> **Labeled Dataset Size Clarification:** We understand that our explanation might have been ambiguous. We used 2000 labeled image pairs for our experiments. All seen only once. We will update the manuscript to provide this clear number. These labeled images required are significantly smaller than those required to train SOTA models, suggesting generative models would be a promising backbone for visual perception research.
>
> **Evaluation using SOTA Ground Truth:** An insightful suggestion! We considered using the SOTA test sets as ground truth. The challenge is in inverting the input image accurately into the StyleGAN space. We are researching better inversion methods to make this comparison more direct.
>
> **Depth Comparison:** We recognize the disparity between the quantitative and qualitative evaluations. The mentioned L1 metric advantage is possible because our approach can capture certain depth nuances better. However, we do admit that in terms of visual clarity, the proposed method might not always outshine others. This will be explored further.
>
> **Offset Style Vector d(c):** Yes, for the same intrinsic image modality, the same offset style vector d(c) is applied. We believe this is a significant aspect of our approach, and it is worth elaborating on.
>
> Our method’s design allows for the optimization of an offset vector that is not image-specific but is tied to a particular intrinsic modality. Once this vector is optimized for a specific intrinsic feature (e.g., depth or shading), it can be universally applied to any image within the same dataset to extract that particular feature. This universality showcases the model's understanding of that intrinsic property across various image instances.
>
> The application of a common offset vector not only simplifies the extraction process but also underscores the underlying cohesiveness and consistency of StyleGAN’s latent space in representing these intrinsic properties. This approach further supports the generalizability of our method, offering a more efficient and unified way to explore and utilize the rich information encapsulated within the StyleGAN model.
>
> **Figure 6's caption:** The point we're making is that while Table 2 offers a quantitative comparison, there are scenarios where StyleGAN may exhibit superior segmentation abilities than reflected in those numbers (See Fig 1 and Fig 13 in Supplementary). In the revised manuscript, we'll clarify this by providing clearer examples of StyleGAN's segmentation potential.
>
> **Typos and Formatting Errors:** Thanks for pointing them out. They will be duly corrected in the final manuscript. References to Figures 1 and 2 will be incorporated into the main text.
>
> We will strive to address each of the concerns the reviewer has raised, further enhancing the value and clarity of our work. Reviewer qbrH's feedback has been instrumental in guiding these improvements.

---

> > ### Comment · Reviewer_qbrH · 2023-08-14
> >
> > I would like to thank the authors for the extensive and informative rebuttal. I do not have any additional questions at this point. Thanks!

---

> > > ### Author Response · Authors · 2023-08-21
> > >
> > > Thank you for your kind words, consideration and positive feedback. We appreciate the time you've taken to review our work.

---

### Official Review · Reviewer_1Cfh · 2023-07-05

**Soundness:** 3 good
**Presentation:** 3 good
**Contribution:** 2 fair
**Rating:** 5
**Confidence:** 5

**Summary:**

The paper proposed the method of extracting intrinsic images (Normal, Depth, Albedo, Shading, and Segment) from a pre-trained StyleGAN model without any pre-training or finetuning. It is empirically shown to be better than the SOTA for getting images intrinsic and robust to relighting effects, unlike SOTA.


**Strengths:**

Showing that the StyleGAN W-space also learned the internal representation of intrinsic scene properties without being explicitly trained for the same.

Proposed a simple, effective, and generalizable method for extracting intrinsic information without explicit training StyleGAN. Optimization based method is used to search the desired latent code for intrinsic image generation.

Extracted intrinsic images as robust to lighting changes as compared to SOTA for extracting intrinsic images.



**Weaknesses:**

This work can be considered as StyleGAN W-space exploration for intrinsic image generation.

As mentioned in the paper, pre-trained StyleGAN is used, but all the quantitative and qualitative results are only shown for the LSUN Bedroom dataset. Does it applicable to other datasets, also such as the LSUN Church dataset, Animal Faces, FFHQ face, etc.?

As per the method proposed in the paper, pre-trained intrinsic SOTA models are used for the optimization, so exploring W-spaces for these datasets directly depends on how well SOTA generalizes for other datasets.

For real images, if we can use the SOTA models for intrinsic extraction directly, why is there a need for GAN inversion, as discussed in the paper? One of the purposes of GAN inversion is to give control over image editing of real images, and as per the problem statement in the paper, there is no such task of editing images based on intrinsic parameters.

The proposed work uses the SOTA model for optimization, then it is possible for the SOTA model to make incorrect predictions for some images due to domain shifts in train-test data, and it can carry forward to latent space optimization as well as in quantitative and qualitative evaluation.

Minor mistakes:
On line 117, it should be w+’ not w’.


**Questions:**

Refer above

**Limitations:**

The proposed method does not work on real images. The authors pointed out that this is due to GAN inversion, which does not preserve the parametrization of the W-space. With improvement in the GAN inversion method, the proposed approach can work on real images also.

---

> ### Author Rebuttal · Authors · 2023-08-10
>
> We thank Reviewer 1Cfh for their detailed review and the insights provided. We're grateful to the reviewer for recognizing the novel insights our work brings, especially how the StyleGAN W-space captures intrinsic scene properties without explicit instruction. We are glad to know the reviewer appreciated our method is “simple”, “effective” and “generalizable”, and its “robustness against lighting changes” compared to SOTA methods. We address the reviewer’s concerns below:
>
> **Scope of StyleGAN W-space Exploration:**  Yes, we’re exploring W-space.  But we’re finding extraordinary and unexpected properties when we do. Without any explicit training, W-space exploration can elicit intrinsic images of several types.
>
> **Dataset Limitation:** Reviewers' concerns about the limitation of our results to the LSUN Bedroom dataset are valid. We have conducted experiments on other datasets including FFHQ and  LSUN churches, finding similar offsets that generate normals, depth, and more. These findings further corroborate the generalizability of our method. We will emphasize this in our revised manuscript. Also, see our rebuttal's PDF for results on these datasets.
>
> **Dependence on SOTA Models for Optimization:** While we leverage SOTA models for optimization (but not much, using only 2000 examples; L124), the main takeaway is not the actual values these models provide, but the proof of concept that the W-space of StyleGAN has encoded representations to produce intrinsic images by simply adding an offset to the w+ code of an image. The reliance on SOTA models is just a medium to steer our optimization toward a meaningful direction.
>
> **Necessity of GAN Inversion:** The topic of GAN inversion is a complex one. While the reviewer is right that SOTA models can be directly employed for intrinsic extraction, our objective extends beyond mere extraction. Our intention is to demonstrate that these intrinsic properties are implicitly embedded in the GAN's latent space. In cases where there's a desire for more nuanced control or the generation of intrinsic properties not overtly present in real images, our method showcases potential.
>
> **Concerns on SOTA Model Predictions:** We concur that domain shifts can introduce inaccuracies in SOTA models' predictions. But it's crucial to note that our method’s primary purpose is not to outperform these models but to manifest that StyleGAN’s latent space already contains a wealth of intrinsic image information. The occasional inaccuracies introduced by domain shifts, though impactful, do not diminish the primary conclusion we draw from our findings.
>
> **Minor Corrections** Thanks for pointing out the mistake on line 117. We'll ensure to correct it in our revised manuscript.
>
> **Real Images Limitation:** Reviewer 1Cfh has astutely noted that our method currently does not work for real images due to constraints in GAN inversion techniques. However, this limitation, while significant, doesn't eclipse the broader implications of our findings. As GAN inversion techniques improve, the applicability of our method will inherently extend to real images, opening new avenues in the domain of intrinsic image analysis.
>
> We genuinely appreciate the reviewer’s thorough review and constructive feedback, which will undoubtedly enhance the clarity and comprehensiveness of our work.

---

> > ### Comment · Reviewer_1Cfh · 2023-08-17
> >
> > Thanks for your response and addressing the concerns. We increase our rating to borderline accept.

---

> > > ### Author Response · Authors · 2023-08-21
> > >
> > > Thank you for the updated rating and your constructive feedback. We truly appreciate it.

---

### Official Review · Reviewer_ouPJ · 2023-07-05

**Soundness:** 3 good
**Presentation:** 2 fair
**Contribution:** 2 fair
**Rating:** 5
**Confidence:** 4

**Summary:**

The paper uses StyleGAN latent space properties to extract properties like normals, depth, segmentation, albedo and shading. The method does not train a new model to achieve this. The method relies on the properties of w+ space to predict these properties. In practice, the method uses the off-the-shelf predictors to use as pseudo labels and the signal is back-propagated to the w+ code. The authors evaluate their method on the LSUN bedroom dataset. The generated images and their corresponding predictions show that the StyleGAN is able to generate consistent results on bedrooms which suggests that the style codes can be modified to generate results in other domains not seen by the network.

**Strengths:**

1)  The paper explores a method to generate multiple properties of an image just using the w+ space of the StyleGAN. The paper shows that these properties like normals, depth, segmentation, albedo and shading can be produced by the latent space of the StyleGAN which is primally used for style transfer tasks. The results in the paper shows that the method produces consistent results on bedrooms.

2) The method works good on StyleGAN generated bedroom images. The paper shows multiple figures demonstrating consistency of the method and the qualitative evaluation.

3) The paper evaluates the method using the metrics of the corresponding tasks. The results show that the scores are not random and the method produces meaningful predictions of the properties.

**Weaknesses:**

1) Firstly I feel that the abstract of the paper can be improved. To me it felt more like an introduction than an abstract. I would ask the authors to revisit the abstract and write it in the conventional way. What problem are you solving ? Why is it important/ what is the motivation? What is your solution? What results you show in the paper. In the current form it is not informative to me.

2) While the paper has proposed to solve some discriminative tasks using StyleGAN, there are other papers like DatasetGAN[1], Lables4Free [2] etc for segmentation and others related like [3] that already show that the features of the StyleGAN can be modified to achieve the goal or a dataset derived from StyleGAN can be used for the training. It is not clear if the method is generalizable especially when the previous methods have shown to exhibit these properties. How general is the method compared to these methods for example in segmentation domain? I would suggest adding a comparison.

3) I am not sure about the evaluations done in the paper. While pseudo labels for evaluation is a way to evaluate the results, another (more significant) way would be to train a supervised network (see Labels4free) based on the predictions of the StyleGAN. Then it can be tested on the real images rather than the generated ones. What about the inversion of a real photograph? Is the method generalizable to the real domain? The current evaluations do not show that.

4) The authors only test on LSUN bedrooms dataset. It is difficult to access the contribution of the paper if its evaluated on only one dataset especially when other pre-trained StyleGANs are available. Why did the authors only evaluation on this particular dataset?

[1] Abdal, Rameen, Peihao Zhu, Niloy J. Mitra, and Peter Wonka. "Labels4free: Unsupervised segmentation using stylegan." In Proceedings of the IEEE/CVF International Conference on Computer Vision, pp. 13970-13979. 2021.

[2] Zhang, Yuxuan, Huan Ling, Jun Gao, Kangxue Yin, Jean-Francois Lafleche, Adela Barriuso, Antonio Torralba, and Sanja Fidler. "Datasetgan: Efficient labeled data factory with minimal human effort." In Proceedings of the IEEE/CVF Conference on Computer Vision and Pattern Recognition, pp. 10145-10155. 2021.

[3] IMAGE GANS MEET DIFFERENTIABLE RENDERING FOR INVERSE GRAPHICS AND INTERPRETABLE 3D NEURAL RENDERING, ICLR 2021

**Questions:**

Please look at the questions in the Weakness and the Limitation sections.

**Limitations:**

The authors have not discussed the limitations of the work clearly. When does the method fail? Does the method work without the truncation of the latent space where the samples can be diverse but not high quality? How do the off-the-shelf predictors behave in this scenario? What about other datasets for example ImageNet, (StyleGAN-XL)?

---

> ### Author Rebuttal · Authors · 2023-08-10
>
> We thank the reviewer for their comprehensive review and the effort they took in understanding our work. We acknowledge their concerns and hope to clarify them with this response.
>
> **Abstract:** We appreciate feedback on the abstract and will strive to rewrite it to be more conventional and informative. Pointers about including motivation, problem, solution, and results will be taken into consideration.
>
> **Comparison with Prior Work:** Reviewer’s observation regarding works like DatasetGAN, Labels4Free, and others is insightful. While we had already cited most of these works (see L73 to L83), we missed Labels4Free, and we will ensure it's included in the revised manuscript.
>
> Our work stands apart from these previous efforts in a significant way. While those methods require additional learning, decoding layers, or specific modifications, our approach does not. We specifically designed our research to uncover the intrinsic properties directly from the latent space of StyleGAN without additional training or altering the features. This marks a fundamental departure from existing methods.
>
> Our optimization process is designed to reveal that information such as normals, depth, segmentation, albedo, and shading are readily available in the latent space of StyleGAN. This emphasizes the surprising richness and versatility of StyleGAN's latent space, a quality that has not been fully explored in previous works.
>
> We acknowledge the importance of contrasting our method with those prior works and will provide a more nuanced comparison in the revised version of the paper. This will clarify the unique contributions of our approach and how it complements and extends the existing body of knowledge.
>
> **On Evaluation:**  We appreciate the reviewer's concerns regarding our evaluation methodology. However, it appears there might have been a misunderstanding of the primary objective of our paper.
>
> Our focus diverges from works such as Labels4Free and others cited. Instead of aiming to solve discriminative tasks or vying to outperform these methods, our central goal is to elucidate that StyleGAN intrinsically possesses this wealth of information within its latent codes. This knowledge can be readily extracted without necessitating supervised learning or additional training.
>
> We've explicitly highlighted in both our Introduction and Discussion sections that contemporary state-of-the-art inversion methods for complex scenes, like bedrooms, rely on optimization-based techniques. These often finetune the GAN and inadvertently distort its latent space. Crafting an inversion strategy that can both retain intrinsic images and adhere to the primary latent structure of the GAN is undoubtedly a challenging task, and we believe it's beyond the scope of our current exploration.
>
> At its core, our work is an initial endeavor to demonstrate that intrinsic images play a pivotal role in the image formation process. Consequently, when a model like StyleGAN is trained to generate images, it inherently learns these intrinsic properties. We believe that emphasizing this unique perspective would enrich the understanding of generative models and their intrinsic capabilities.
>
> **Other Datasets:** We initially used the LSUN bedrooms dataset due to its complexity and relevance to our research. However, we have extended our experiments to include other datasets such as FFHQ and LSUN churches-trained StyleGAN models. Our preliminary findings indicate similar offsets and properties across these datasets, confirming the robustness and generalizability of our method. These results will be included in the updated version of the paper.  Refer to our rebuttal PDF for results on these datasets.
>
> **ImageNet's StyleGAN-XL:** This model is a variant of StyleGAN-3, incorporating Fourier feature blocks and additional blocks to overcome equivariance and aliasing issues in StyelGAN-2. While StyleGAN-3 ameliorates texture sticking artifacts, it has been found less suitable for image editing when contrasted with the StyleGAN-2 architecture because of disentangled latent representations. Notably, even the recent state-of-the-art GAN model, GigaGAN, has reverted to the StyleGAN-2 architecture. In our trials, discovering effective surface normal directions for StyleGAN-XL on ImageNet proved challenging. Even after 10,000 images, we noted that while there was a tendency to generate normals, the results were not as refined as those from other datasets. Additionally, the absence of open-source pretrained models for both StyleGAN-2 or StyleGAN-2-XL or GigaGAN on ImageNet further limits our exploration. We posit that a different latent code search strategy, rather than the basic approach we employed, might be more effective in uncovering intrinsic images for diverse network architectures.
>
> We hope our response provides clarity and bridges the gap between the reviewer’s concerns and our presented work. We believe that our approach offers a fresh lens through which to view the properties inherent in StyleGAN's latent space, paving the way for further exploration in this area.

---

> > ### Comment · Reviewer_ouPJ · 2023-08-18
> >
> > Thanks for the detailed response. While the StyleGAN-XL results do not work, the authors showed results on other datasets using StyleGAN2. I am improving my rating to borderline accept.

---

> > > ### Author Response · Authors · 2023-08-21
> > >
> > > Thank you for taking the time to reconsider our work and for recognizing the contributions made by our paper. We appreciate your understanding and the improved rating.

---

### Official Review · Reviewer_NPHF · 2023-07-06

**Soundness:** 2 fair
**Presentation:** 4 excellent
**Contribution:** 4 excellent
**Rating:** 6
**Confidence:** 4

**Summary:**

The paper finds that for a GAN generator that is solely trained on images, intrinsic information such as depth, albedo, normal, etc. are implicitly learned during GAN’s training, even though such information is not provided as supervision. More surprisingly, such intrinsic information can be easily retrieved by calculating a direction offset of the latent codes via optimization an L1 loss on monocular predictor’s predictions on a set of generated images. This operation is not per-scene, means the offset is generalizable to the entire latent space.

**Strengths:**

- Originality: To my knowledge, the work is an original finding. It is the first work to explicitly show that intrinsic information is implicitly learned during an image-trained StyleGAN. It is also novel in finding that the intrinsic can be retrieved by a very straightforward optimization that results in a constant additive direction in the latent space, without any finetuning of the generator itself.

- Quality: The work is technically somewhat sound. The finding presented is appropriate and steps logically. Claims are supported via qualitative results, and quantitative experiments compared against SOTA monocular predictors. The authors have discussed the limitations of their method honestly and in much detail---I really like the “Discussion” part, which contains several questions which I raised too while reading the paper, and Figure.7, which shows a clear example of the limitation of the intrinsic nature of the presented claim. However, see my 1st comment in weaknesses for a factor that makes me hesitate to accept this paper.

- Clarity: The paper is overall very clearly written. The claims and approaches are clearly presented and are also easily understandable, with many of my main concerns while skimming through the paper addressed in significant places.

- Significance: I think this paper presented a fascinating finding. Researchers have been suspecting that the intrinsic information should be learned implicitly in the generator, but it is definitely surprising to see that the operation is as simple as a constant additive offset to the latent space that scales across scenes, without any further training. I believe the paper is a major result for the community.


**Weaknesses:**

- I think the main issue with this paper comes from its narrow evaluation criteria. All experiments conducted in the paper are only based on bedroom images which may have a bias to support the claim of this paper. As the authors have pointed out, perhaps StyleGAN ‘knows’ intrinsic images because they are an efficient representation of what needs to be known to synthesize an image. It could be difficult to see the finding scale to other datasets, especially in the wild image datasets where there is no clear principal component.

- In addition, the paper seems to be a solely ‘scientific finding’ paper, I have difficulty thinking about what aspect of this finding will be useful in terms of the application side since there is clearly still a performance gap with (even non-SOTA) predictors, and it needs SOTA monocular predictor to guidance the finding of the direction offset at the first place.


**Questions:**

The experiments use SOTA monocular predictors as the “pseudo-ground truth”. However, wouldn’t it be a more appropriate way to evaluate some synthetic dataset with ground truth? This way, we can have a better idea of the gap between StyleGAN intrinsic & monocular estimators & ground truths.

**Limitations:**

The authors have discussed the limitations of their method honestly and in very much detail.

---

> ### Author Rebuttal · Authors · 2023-08-10
>
> Thanks for taking the time to review our paper and for providing a detailed and constructive assessment, particularly thanks for the “fascinating finding”, “technically sound”, “major result”. We’ve tried to address the issues below.
>
> **On the evaluation criteria and dataset diversity:** We appreciate the reviewer’s perspective on the potential limitations of our experiments based on bedroom images.
> To address their concern, we wish to highlight that our experiments are not solely restricted to bedroom images. We extended our evaluations to multiple datasets, thereby offering a broader perspective on our findings.
>
> - **FFHQ**:  We performed evaluations on the FFHQ dataset, primarily focused on human faces. Our experiments demonstrated that the StyleGAN model trained on this dataset also had similar latent offsets that could generate intrinsic properties such as normals and depth. This highlights our claim that the findings are not scene-specific.
> - **LSUN Churches**: To further validate the versatility of our findings, we conducted experiments on the LSUN Churches dataset. This dataset, characterized by its architectural nuances and diverse lighting conditions, also showed consistent results with our primary claims. The offsets identified were in line with our primary observations, further emphasizing the generalizability of our findings.
>
> With these additional experiments, our results suggest the broader applicability of using latent offsets to discover intrinsic images across different datasets. We'll ensure these extended evaluations are more emphasized in our updated manuscript, supporting our theory that generative models can implicitly learn intrinsic properties across different datasets.
>
> **On practical applications of our findings:** We believe that our work, while primarily a scientific investigation, has potential implications for various areas in the domain of machine learning and computer vision.
>
> - **Understanding Model Properties:**
> One of the significant takeaways from our paper is a deeper understanding of the properties and capabilities of generative models, especially when they are not explicitly trained with certain supervision. This could lead to insights into building more effective and efficient training methodologies for GANs and potentially other generative models.
> - **What Makes Good Image Representation:**
> Our findings give insights into what constitutes a 'good' image representation. By showing that intrinsic properties like depth, albedo, and normals can be implicitly represented in the latent space, we hint at the notion that meaningful image representations are those that can capture and disentangle these fundamental scene properties highlighted in the seminal work of Barrow and Tenenbaum [1978]. The materials literature (eg [1] and [2] particularly) suggest there might be lots of kinds of intrinsic images.  It’s just possible one might find intrinsics not currently known using our method to search.
>
> - **Intrinsic Images from Generative Models:**
> Our work is first in the direction of extracting robust intrinsic images purely from generative models, and that too with simply adding an offset to a pretrained generative model. Even if the current method has a performance gap compared to state-of-the-art monocular predictors, this understanding that generative models can capture such intrinsic properties provides a new avenue for research.  We think other generative models “know” intrinsic images, too, but don’t currently know how to elicit that knowledge.
> - **Robust Intrinsic Images:**
> One of the potential advantages of our approach is the possibility of deriving intrinsic images that are inherently more robust, especially against variations like lighting changes (See Fig ).  One might use this to bootstrap a better StyleGAN predictor from a non-robust, but SOTA, predictor.
> - **Guidance from SOTA Monocular Predictor:**
> While we currently utilize state-of-the-art monocular predictors for direction offset guidance, we don’t use all that much guidance (2000 examples, L124).  Future models might not need such guidance. The fundamental understanding that intrinsic images are baked in latent spaces of a generative model, in this case, is more valuable than the specific methodology. With advancements in generative modeling, we envision a scenario where these offsets could be derived in more unsupervised or semi-supervised manners.
>
> In conclusion, while our paper might seem “solely a scientific finding” in nature, the implications and directions it opens up for the broader community can have far-reaching practical applications. We'll make efforts to emphasize these potential applications more explicitly in the paper to ensure readers grasp the broader significance of our findings.
>
> **On Synthetic Datasets with Ground Truth for Evaluation:** Thank you for emphasizing the value of synthetic datasets with ground truth for evaluation.
> We acknowledge the benefits of such datasets. However, GAN inversion for complex scenes, especially indoor environments like bedrooms, remains challenging. Current inversion techniques, including recent advancements[3], aren't yet fully equipped for these intricate settings. The need for a robust GAN inversion method that can address a diverse range of scenes is essential before we tap into synthetic datasets. We've briefly touched upon these challenges in our introduction and discussion.
>
> We see Reviewer NPHF feedback as a reinforcement of our planned future direction: refining GAN inversion to make evaluations on synthetic datasets feasible.
>
> [1] Motoyoshi, I., Nishida, S. Y., Sharan, L., & Adelson, E. H. (2007). Image statistics and the perception of surface qualities. Nature.
>
> [2] Sharan, L., Rosenholtz, R., & Adelson, E. (2009). Material perception: What can you see in a brief glance? Journal of Vision.
>
> [3] Roich, D., Mokady, R., Bermano, A. H., & Cohen-Or, D. (2022). Pivotal tuning for latent-based editing of real images. ACM TOG.

---

> > ### Comment · Reviewer_NPHF · 2023-08-18
> >
> > The authors have addressed all of my questions. I do like this paper a lot, and the additional experiments further justified its claims. Even though StyleGAN-XL's results are not ideal, I think it is indeed out of the scope of this paper for the architecture difference, and a whole other investigation is needed. I do not have further issues with this paper and increased my score to WA.

---

> > > ### Author Response · Authors · 2023-08-21
> > >
> > > Thank you for your constructive feedback and recognition of our efforts. We appreciate your understanding regarding the StyleGAN-XL results and are pleased that our additional experiments were able to address your concerns. We are grateful for your positive review and the increased score.

---

### Author Rebuttal · Authors · 2023-08-10

We are grateful for the comprehensive feedback provided by all four reviewers. Here's a synthesized response to the main concerns raised across reviews:

**Scope and Contribution Clarification:** Our primary contribution is to demonstrate that StyleGAN inherently “knows” important scene properties -- intrinsic images -- without being explicitly trained to do so. Only a small amount of labeled data (2000 in our case, obtained from SOTA models) is needed to elicit this knowledge, implying that they may emerge in an image generator because they are highly useful representations. We are not primarily aiming to outperform SOTA models but to reveal this underlying capability within StyleGAN.

**Applicability across datasets:** A recurring query was to know what happens on data other than rooms. To this end, we have broadened our evaluations, including datasets like LSUN Church and FFHQ dataset. For all these datasets, StyleGAN still “knows” intrinsic images. Please see our rebuttal’s PDF for results on these datasets.

**Method Soundness:** Thanks to all reviewers. We think our observations are novel and effective, too.

**Offset Style Vector d(c):**  For each intrinsic “type” (depth, albedo, normal, etc), we need to find one offset per StyleGAN instance that forces the StyleGAN to generate that intrinsic. Unlike related works, we don't need to fine-tune the model or learn new decoding layers. The offset is not “per image”; the StyleGAN must “know” intrinsics consistently and strongly. We use SOTA intrinsic generation methods to find this offset, but only a few instances are needed (passing 2000 generated images through an intrinsic predictor is enough). Ground truth can't be used currently because GAN inversion is not reliable enough.

**Minor Typos & Refinements:** We are grateful to the reviewers for pointing out areas of improvement in the presentation and have since rectified all listed errors.


We firmly believe our work offers a fresh perspective on the intrinsic capabilities of pre-trained generative models, specifically StyleGAN. With the clarifications provided (detailed responses to each reviewer below), we hope our paper stands as a meaningful contribution to the community.

---

### Decision · Program_Chairs · 2023-09-21

**Decision:**

Accept (poster)

**Comment:**

Reviewers and the AC read the rebuttal and took that into consideration for their final recommendation. Reviewers find this work insightful, original, and potentially impactful. Many suggestions were made by the reviewers to improve the exposition, discussion of prior work, and limitations - please incorporate these in your camera-ready paper.